# Clinical characteristics, racial inequities, and outcomes in patients with breast cancer and COVID-19: A COVID-19 and cancer consortium (CCC19) cohort study

Gayathri Nagaraj[1]*[†], Shaveta Vinayak[2,3,4†], Ali Raza Khaki[5†], Tianyi Sun[6], Nicole M Kuderer[3,7], David M Aboulafia[8], Jared D Acoba[9], Joy Awosika[10], Ziad Bakouny[11], Nicole B Balmaceda[12], Ting Bao[13], Babar Bashir[14], Stephanie Berg[15], Mehmet A Bilen[16], Poorva Bindal[17], Sibel Blau[18], Brianne E Bodin[19], Hala T Borno[20], Cecilia Castellano[16], Horyun Choi[9], John Deeken[21], Aakash Desai[22], Natasha Edwin[23], Lawrence E Feldman[24], Daniel B Flora[25], Christopher R Friese[26], Matthew D Galsky[27], Cyndi J Gonzalez[26], Petros Grivas[2,3,4], Shilpa Gupta[28], Marcy Haynam[29], Hannah Heilman[10], Dawn L Hershman[19], Clara Hwang[30], Chinmay Jani[31], Sachin R Jhawar[29], Monika Joshi[32], Virginia Kaklamani[33], Elizabeth J Klein[34], Natalie Knox[35], Vadim S Koshkin[20], Amit A Kulkarni[36], Daniel H Kwon[20], Chris Labaki[11], Philip E Lammers[37], Kate I Lathrop[33], Mark A Lewis[38], Xuanyi Li[6], Gilbert de Lima Lopes[39], Gary H Lyman[2,3,4], Della F Makower[40], Abdul-Hai Mansoor[41], Merry-Jennifer Markham[42], Sandeep H Mashru[41], Rana R McKay[43], Ian Messing[44], Vasil Mico[14], Rajani Nadkarni[45], Swathi Namburi[18], Ryan H Nguyen[24], Taylor Kristian Nonato[43], Tracey Lynn O'Connor[46], Orestis A Panagiotou[34], Kyu Park[1], Jaymin M Patel[17], Kanishka GopikaBimal Patel[47], Jeffrey Peppercorn[48], Hyma Polimera[32], Matthew Puc[49], Yuan James Rao[44], Pedram Razavi[43], Sonya A Reid[6], Jonathan W Riess[47], Donna R Rivera[50], Mark Robson[13], Suzanne J Rose[51], Atlantis D Russ[42], Lidia Schapira[5], Pankil K Shah[33], M Kelly Shanahan[52], Lauren C Shapiro[40], Melissa Smits[23], Daniel G Stover[29], Mitrianna Streckfuss[53], Lisa Tachiki[2,3,4], Michael A Thompson[53], Sara M Tolaney[11], Lisa B Weissmann[31], Grace Wilson[36], Michael T Wotman[27], Elizabeth M Wulff-Burchfield[12], Sanjay Mishra[6], Benjamin French[6], Jeremy L Warner[6], Maryam B Lustberg[54‡], Melissa K Accordino[19‡], Dimpy P Shah[33]*[‡], On behalf of the COVID-19 and Cancer Consortium

*For correspondence:
gnagaraj@llu.edu (GN);
shahdp@uthscsa.edu (DPS)

[†]Co-Primary authors

[‡]Co-Senior authors

Competing interest: The authors declare that no competing interests exist.

[1]Loma Linda University Cancer Center, Loma Linda, United States; [2]Fred Hutchinson Cancer Research Center, Seattle, United States; [3]University of Washington, Seattle, United States; [4]Seattle Cancer Care Alliance, Seattle, United States; [5]Stanford University, Palo Alto, United States; [6]Vanderbilt University Medical Center, Nashville, United States; [7]Advanced Cancer Research Group, Kirkland, United States; [8]Virginia Mason Cancer Institute, Seattle, United States; [9]University of Hawaii Cancer Center, Honolulu, United States; [10]University of Cincinnati Cancer Center, Cincinnati, United States; [11]Dana-Farber Cancer Institute, Boston, United States; [12]The University of Kansas Cancer Center, Kansas City, United States; [13]Memorial Sloan Kettering Cancer Center, New York, United States; [14]Sidney Kimmel Comprehensive Cancer Center, Thomas Jefferson University, Philadelphia, United States; [15]Loyola University Medical Center, Maywood, United States; [16]Winship Cancer Institute, Emory University,

Atlanta, United States; [17]Beth Israel Deaconess Medical Center, Boston, United States; [18]Northwest Medical Specialties, Tacoma, United States; [19]Herbert Irving Comprehensive Cancer Center, Columbia University, New York, United States; [20]Helen Diller Family Comprehensive Cancer Center, University of California, San Francisco, San Francisco, United States; [21]Inova Schar Cancer Institute, Fairfax, United States; [22]Mayo Clinic, Rochester, United States; [23]ThedaCare Cancer Care, Appleton, United States; [24]University of Illinois Hospital & Health Sciences System, Chicago, United States; [25]St. Elizabeth Healthcare, Edgewood, United States; [26]Rogel Cancer Center, University of Michigan-Ann Arbor, Ann Arbor, United States; [27]Tisch Cancer Institute, Icahn School of Medicine at Mount Sinai, New York, United States; [28]Cleveland Clinic, Cleveland, United States; [29]The Ohio State University Comprehensive Cancer Center, Columbus, United States; [30]Henry Ford Cancer Institute, Henry Ford Hospital, Detroit, United States; [31]Mount Auburn Hospital, Cambridge, United States; [32]Penn State Health St Joseph Cancer Center, Reading, United States; [33]Mays Cancer Center, The University of Texas Health San Antonio MD Anderson Cancer Center, San Antonio, United States; [34]Lifespan Cancer Institute, Brown University, Providence, United States; [35]Stritch School of Medicine, Loyola University, Maywood, United States; [36]Masonic Cancer Center, University of Minnesota, Minneapolis, United States; [37]Baptist Cancer Center, Memphis, United States; [38]Intermountain Healthcare, Salt Lake City, United States; [39]Sylvester Comprehensive Cancer Center, University of Miami Miller School of Medicine, Miami, United States; [40]Montefiore Medical Center, Albert Einstein College of Medicine, Bronx, United States; [41]Kaiser Permanente Northwest, Portland, United States; [42]Division of Hematology and Oncology, University of Florida Health Cancer Center, Gainesville, United States; [43]Moores Cancer Center, University of California, San Diego, San Diego, United States; [44]Division of Radiation Oncology, George Washington University, Washington, United States; [45]Hartford HealthCare Cancer Institute, Hartford, United States; [46]Roswell Park Comprehensive Cancer Center, Buffalo, United States; [47]UC Davis Comprehensive Cancer Center, University of California, Davis, Davis, United States; [48]Massachusetts General Hospital, Boston, United States; [49]Virtua Health, Marlton, United States; [50]Division of Cancer Control and Population Sciences, National Cancer Institute, Rockville, United States; [51]Carl & Dorothy Bennett Cancer Center, Stamford Hospital, Stamford, United States; [52]METAvivor, Annapolis, United States; [53]Aurora Cancer Care, Advocate Aurora Health, Milwaukee, United States; [54]Yale Cancer Center, Yale University School of Medicine, New Haven, United States

## Abstract

**Background:** Limited information is available for patients with breast cancer (BC) and coronavirus disease 2019 (COVID-19), especially among underrepresented racial/ethnic populations.

**Methods:** This is a COVID-19 and Cancer Consortium (CCC19) registry-based retrospective cohort study of females with active or history of BC and laboratory-confirmed severe acute respiratory syndrome coronavirus-2 (SARS-CoV-2) infection diagnosed between March 2020 and June 2021 in the US. Primary outcome was COVID-19 severity measured on a five-level ordinal scale, including none of the following complications, hospitalization, intensive care unit admission, mechanical ventilation, and all-cause mortality. Multivariable ordinal logistic regression model identified characteristics associated with COVID-19 severity.

**Results:** 1383 female patient records with BC and COVID-19 were included in the analysis, the median age was 61 years, and median follow-up was 90 days. Multivariable analysis revealed higher odds of COVID-19 severity for older age (aOR per decade, 1.48 [95% CI, 1.32–1.67]); Black

patients (aOR 1.74; 95 CI 1.24–2.45), Asian Americans and Pacific Islander patients (aOR 3.40; 95 CI 1.70–6.79) and Other (aOR 2.97; 95 CI 1.71–5.17) racial/ethnic groups; worse ECOG performance status (ECOG PS ≥2: aOR, 7.78 [95% CI, 4.83–12.5]); pre-existing cardiovascular (aOR, 2.26 [95% CI, 1.63–3.15])/pulmonary comorbidities (aOR, 1.65 [95% CI, 1.20–2.29]); diabetes mellitus (aOR, 2.25 [95% CI, 1.66–3.04]); and active and progressing cancer (aOR, 12.5 [95% CI, 6.89–22.6]). Hispanic ethnicity, timing, and type of anti-cancer therapy modalities were not significantly associated with worse COVID-19 outcomes. The total all-cause mortality and hospitalization rate for the entire cohort was 9% and 37%, respectively however, it varied according to the BC disease status.

**Conclusions:** Using one of the largest registries on cancer and COVID-19, we identified patient and BC-related factors associated with worse COVID-19 outcomes. After adjusting for baseline characteristics, underrepresented racial/ethnic patients experienced worse outcomes compared to non-Hispanic White patients.

**Funding:** This study was partly supported by National Cancer Institute grant number P30 CA068485 to Tianyi Sun, Sanjay Mishra, Benjamin French, Jeremy L Warner; P30-CA046592 to Christopher R Friese; P30 CA023100 for Rana R McKay; P30-CA054174 for Pankil K Shah and Dimpy P Shah; KL2 TR002646 for Pankil Shah and the American Cancer Society and Hope Foundation for Cancer Research (MRSG-16-152-01-CCE) and P30-CA054174 for Dimpy P Shah. REDCap is developed and supported by Vanderbilt Institute for Clinical and Translational Research grant support (UL1 TR000445 from NCATS/NIH). The funding sources had no role in the writing of the manuscript or the decision to submit it for publication.

**Clinical trial number:** CCC19 registry is registered on ClinicalTrials.gov, NCT04354701.

---

## Editor's evaluation

These data offer novel and compelling information that could impact treatment decision-making for breast cancer patients, and the development of this registry contributes a valuable resource for future research, including and beyond breast cancer. It is anticipated that this study is the first of multiple publications that leverage this important data infrastructure.

---

## Introduction

The COVID-19 pandemic has had a devastating impact worldwide and within the United States (US) (*World Health Organization, 2021*; *CDC, 2020a*). Previous studies have reported that patients with cancer are at an increased risk for SARS-CoV-2 infection and have higher rates of adverse outcomes with mortality rates ranging from 14% to 33% (*Grivas et al., 2021*; *Garassino et al., 2020*; *Lee et al., 2020*; *Wang et al., 2021*; *de Azambuja et al., 2020*; *Albiges et al., 2020*; *Sharafeldin et al., 2021*; *Lièvre et al., 2020*). COVID-19 has also highlighted the long-standing health inequities in the US, as underrepresented racial and ethnic populations have disproportionately been affected. Some studies have reported non-White race/ethnicity to be an independent risk factor for worse COVID-19 outcomes such as hospitalization and death (*Grivas et al., 2021*; *Wang et al., 2021*; *CDC, 2020b*; *Millett et al., 2020*; *Muñoz-Price et al., 2020*; *Gross et al., 2020*; *Price-Haywood et al., 2020*; *Azar et al., 2020*; *Mackey et al., 2021*; *Garg et al., 2020*; *Mahajan and Larkins-Pettigrew, 2020*; *Kim and Bostwick, 2020*). Recently published data from CCC19 also showed that Black patients with cancer experienced worse COVID-19 outcomes compared to White patients after adjusting for key risk factors including cancer status and comorbidities (*Fu et al., 2022*).

Breast cancer (BC) is the most common cancer diagnosed in females and affects all major racial/ethnic groups (*Siegel et al., 2021*; *Sung et al., 2021*; *SEER, 2021*). There are well-described racial/ethnic differences in BC incidence and outcomes in females in the US attributable to multiple social and biological factors (*Chlebowski et al., 2005*; *Bigby and Holmes, 2005*; *Yedjou et al., 2019*). Few studies have specifically evaluated the impact of COVID-19 in patients with BC; interpretation from prior studies has been limited by small sample sizes (*Vuagnat et al., 2020*; *Kalinsky et al., 2020*). Data specifically on the impact of COVID-19 among underrepresented racial/ethnic groups with BC are also lacking. Understanding the sociodemographic and clinical factors associated with higher risk for adverse COVID-19 outcomes will help guide patient care. Hence, we aimed to evaluate the

prognostic factors, racial disparities, interventions, complications, and outcomes among patients with active or previous history of BC diagnosed with COVID-19.

## Methods

### Study population

The COVID-19 and Cancer Consortium (CCC19) consists of 129 member institutions capturing granular, detailed, and uniform data on demographic and clinical characteristics, treatment information, and outcomes of COVID-19. Details of CCC19 protocol, data collection, and quality assurance have been previously described (*Kuderer et al., 2020*; *COVID-19 and Cancer Consortium. Electronic address: jeremy.warner@vumc.org and COVID-19 and Cancer Consortium, 2020*). This registry-based retrospective cohort study included all female adults (age ≥18 years) with an active or previous history of invasive BC and laboratory-confirmed diagnosis of SARS-CoV-2 by polymerase chain reaction (PCR) and/or serology from March 17, 2020, to June 16, 2021, in the US. Patient records with multiple invasive malignancies including history of multiple invasive BC were excluded; patients with unknown or missing race and ethnicity, inadequate data quality (quality score >4), and those not evaluable for the primary ordinal outcome were also excluded (*supplementary appendix 1*) (*COVID-19 and Cancer Consortium. Electronic address: jeremy.warner@vumc.org and COVID-19 and Cancer Consortium, 2020*). This study was exempt from institutional review board (IRB) review (VUMC IRB#200467) and was approved by IRBs at participating sites per institutional policy. CCC19 registry is registered on ClinicalTrials.gov, NCT04354701.

### Outcome definitions

The primary outcome was a five-level ordinal scale of COVID-19 severity based on each individual patient's most severe reported disease status: none of the following complications; admitted to the hospital; admitted to an intensive care unit (ICU); mechanically ventilated at any time after COVID-19 diagnosis; or death from any cause. Other COVID-19-related complications (cardiovascular; gastrointestinal; and pulmonary complications, acute kidney injury, multisystem organ failure, superimposed infection, sepsis, any bleeding); 30-day mortality; and anti-COVID-19 directed interventions (supplemental oxygen, remdesivir, systemic corticosteroids, hydroxychloroquine, and other treatments) are also reported.

### Covariates

Covariates were selected a priori and included: age; sex; race/ethnicity (non-Hispanic White [NHW], Black, Hispanic, Asian Americans and Pacific Islanders [AAPI], and Other) as recorded in the EHR, based on the Center for Disease Control and Prevention Race and Ethnicity codes (*CDC, 2021*); US census region of reporting institution (Northeast [NE], Midwest [MW], South and West); month/year of COVID-19 diagnosis (classified into 4-month intervals); smoking status; obesity; comorbidities (cardiovascular, pulmonary, renal, or diabetes mellitus); Eastern Cooperative Oncology Group (ECOG) performance status (PS); BC subtypes based on hormone receptor (HR) and human epidermal growth factor receptor 2 (HER2) expression (HR+/HER2-, HR+/HER2+, HR-/HER2+, HR-/HER2- [triple negative], missing/unknown); cancer status at time of COVID-19 diagnosis; timing of most recent anti-cancer therapy relative to COVID-19 diagnosis (never or after COVID-19 diagnosis, 0–4 weeks, 1–3 months, >3 months); and modality of anti-cancer therapy received within 3 months of COVID-19 diagnosis. Cancer status was defined as remission or no evidence of disease (NED) for >5 years, remission or NED for ≤5 years, and active disease, with active disease further classified as responding to therapy, stable, or progressing. Anti-cancer modalities were categorized as chemotherapy; cyclin-dependent kinase (CDK) 4/6 inhibitor; anti-HER2 therapy; other targeted therapy (non-CDK 4/6 inhibitor, non-anti-HER2 therapy); endocrine therapy; immunotherapy; and locoregional therapy (surgery and/or radiation). In the survey, drug classes (modalities) along with a few specific drugs (through checkboxes) were captured. Survey respondents were also encouraged to provide additional details in the free text boxes which were reviewed extensively by the Informatics Core at VUMC, and queries were sent to participating sites to clarify ambiguous reports. CDK 4/6 inhibitor, anti-HER2 therapy, and other targeted therapy information were extracted from free text in the registry survey while the others were checkboxes. In addition, baseline severity of COVID-19 at presentation, classified as

mild (no hospitalization indicated), moderate (hospitalization indicated), and severe (ICU admission indicated), was collected. Other variables included location of patient residence (urban, suburban, rural) and treatment center characteristics (academic medical center, community practice, tertiary care center). The CCC19 data dictionary is available at https://github.com/covidncancer/CCC19_dictionary (*Mishra and Warner, 2023*). The project approved variables used for the analysis are provided in *supplementary appendix 3*.

## Statistical methods

Covariates, outcome definitions, and statistical analysis plan were prespecified by the authors and the CCC19 Research Coordinating Center prior to analysis (*supplementary appendix 2*). Standard descriptive statistics were used to summarize prognostic factors, rates of clinical complications, interventions during hospitalization, and rates of outcomes such as 30-day mortality, hospitalization, oxygen requirement, ICU admission, mechanical ventilation, and overall mortality among racial and ethnic groups. The primary analysis was restricted to females with BC.

Multivariable ordinal logistic regression models for the COVID-19 severity outcome among females with BC included age, race/ethnicity, obesity, ECOG PS, comorbidities, cancer status, anti-cancer therapy and timing, month/year of COVID-19 diagnosis (classified into 4-month intervals), and US census region of reporting institution. These covariates were identified a priori as the most clinically relevant for COVID-19 severity and were included in a single model, given a sufficient number of events and corresponding degrees of freedom. Because the ordinal outcome was assessed over a given patient's total follow-up period, the model included an offset for (log) follow-up time. The results are presented as adjusted odds ratio (ORs) with 95% CIs. Model stability was assessed by comparing unadjusted and adjusted models and variance inflation factors. Graphical methods were used to verify the proportional odds assumption (*Appendix 4—figure 1*). We used the e value to quantify sensitivity to unmeasured confounding for the observed OR for race/ethnicity (*VanderWeele and Ding, 2017*; *Haneuse et al., 2019*). Multiple imputation (20 imputed datasets) was used to impute missing and unknown data for all variables included in the analysis, with some exceptions: unknown ECOG performance score and unknown cancer status were not imputed and treated as a separate category in analyses. Imputation was performed on the largest dataset possible (i.e., after removing test cases and other manual exclusions, but before applying specific exclusion criteria). Analyses were completed using R v4.0.4 (R Foundation for Statistical Computing, Vienna, Austria), including the rms and EValue extension packages. Descriptive statistics for males with BC and females with metastatic BC (MBC) are presented separately but multivariable modeling was not attempted due to small sample sizes.

## Results

### Baseline characteristics and COVID-19 outcomes in female patients with BC

Of the total 12,034 reports on all cancers submitted to the CCC19 registry at the time of this analysis, 1383 females with BC met the eligibility criteria and were included (*Figure 1*). The median age for the cohort was 61 years (IQR 51–72 years) and median follow-up was 90 (IQR 30–135) days. BC subtypes by biomarker distribution in CCC19 registry included: 52% HR+/HER2-, 14% HR+/HER2+, 8% HR-/HER2+, 11% triple negative, and 14% unknown or missing. BC subtype distribution based on biomarkers in the CCC19 cohort are similar to SEER data which adds broader applicability of these findings (*SEER, 2022*). With regard to BC status, 27% were in remission/NED for over 5 years and 32% were in remission/NED for less than 5 years since the initial BC diagnosis and 32% had active cancer (13% had active and responding, 12% had active and stable and 7% had active and progressing cancer). 57% of patients had received some form of anti-cancer therapy within 3 months of COVID-19 diagnosis. The unadjusted total all-cause mortality and hospitalization rate, included in the primary ordinal outcome, for the female cohort was 9% and 37%, respectively. However, the unadjusted rates of COVID-19 outcomes varied by their BC status; females with active and progressing cancer had the highest all-cause mortality (38%) and hospitalization rates (72%) compared to the rest of the group (*Appendix 5—table 1*). Other clinical outcomes for the female cohort included 30-day all-cause mortality (6%), mechanical ventilation (5%), and ICU care (8%). Additional details on patients with BC and COVID-19 by specific characteristics of interest are presented below.

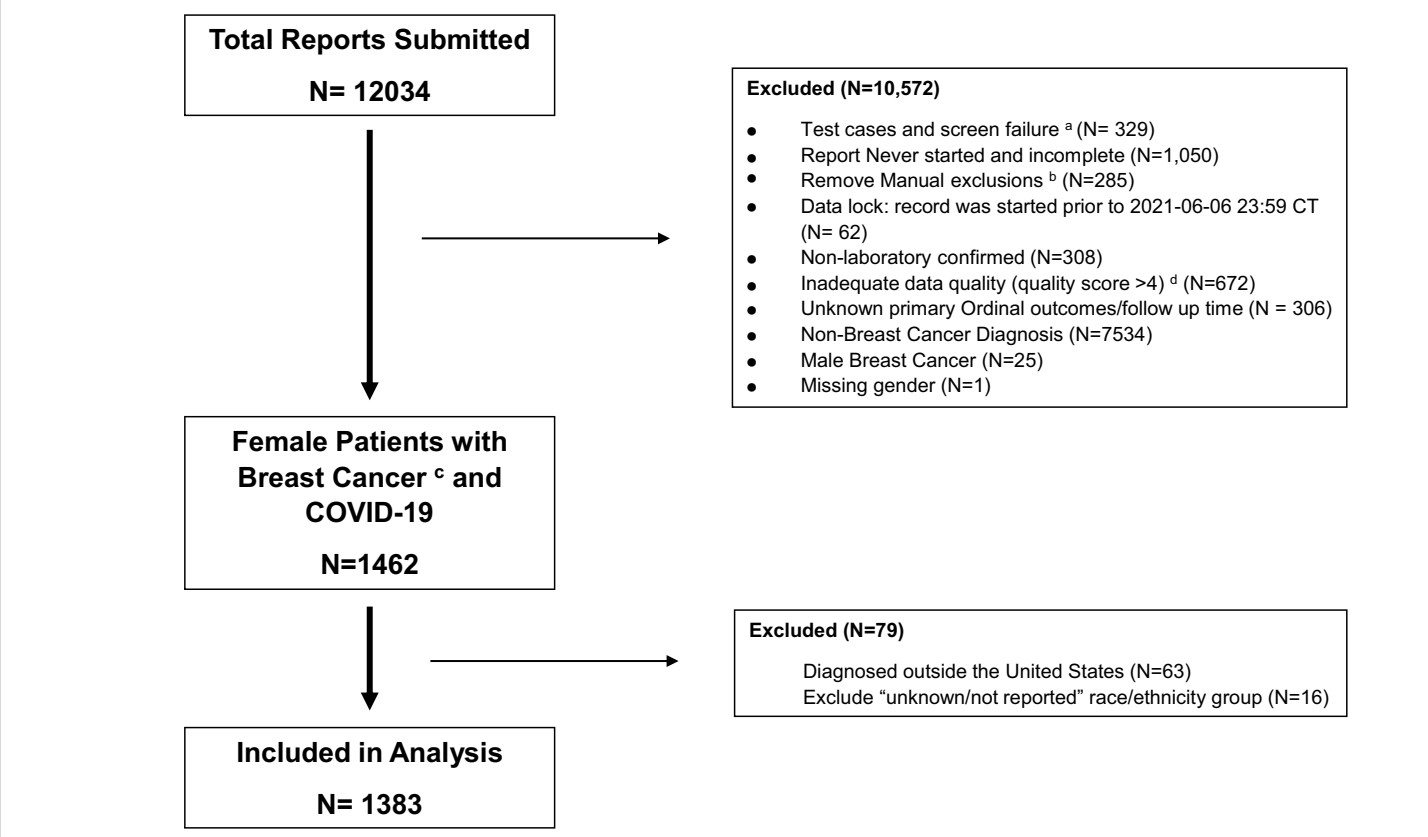

**Figure 1.** Consort flow diagram. Descriptive flow chart of patients included in the study.

## Characteristics of female patients with BC and COVID-19 by race/ethnicity

Of the 1383 female patients, 736 (53%) were NHW, 289 (21%) Black, 235 (17%) Hispanic, 45 (3%) AAPI, and 78 (6%) belonged to Other racial/ethnic group. Baseline characteristics of females stratified by race/ethnicity groups are shown in *Table 1*. Hispanic and AAPI patients were younger with median ages of 53 (IQR 46–62) and 54 (IQR 43–73) years, respectively, compared to 64 years in NHW (IQR 54–76) and 61 years (IQR 52–69) in Black patients. Prevalence of smokers were higher among NHW (35%), Black (33%), and Other (32%) racial/ethnic groups compared to Hispanic (23%) and AAPI (18%) patients. Rates of obesity were higher in Black (54%) and lower in AAPI (29%) compared to NHW (42%) patients. Cardiovascular comorbidity was less common in Hispanic patients (6%), while diabetes mellitus was more prevalent among Black patients (34%) compared to NHW patients (24% and 17%, respectively). Compared to NHW, Hispanic patients had higher rates of active cancer (24% responding, 15% stable, and 9% progressing) and had higher rates of receipt of anti-cancer systemic therapy within 3 months of COVID-19 diagnosis (37% chemotherapy, 25% targeted therapy, 39% endocrine therapy). Similarly, AAPI patients also had higher rates of active cancer (7% responding, 22% stable, and 13% progressing) and received anti-cancer systemic therapy within 3 months of COVID-19 diagnosis (24% chemotherapy, 18% targeted therapy, 33% endocrine therapy) compared to NHW patients with active cancer (9% responding, 12% stable, and 6% progressing) who received anti-cancer systemic therapy (16% chemotherapy, 15% targeted therapy, 38% endocrine therapy).

**Table 1.** Baseline characteristics by race/ethnicity.

| | NHW | Black | Hispanic | AAPI | Others | All |
|---|---|---|---|---|---|---|
| | (n=736, 53%) | (n=289, 21%) | (n=235, 17%) | (n=45, 3%) | (n=78, 6%) | (n=1383, 100%) |
| Median age, years* [IQR] | 64 (54–76) | 61 (52–69) | 53 (46–62) | 54 (43–73) | 62 (53–71) | 61 (51–72) |
| Median follow-up, days [IQR] | 90 (30–135) | 90 (30–180) | 90 (30–135) | 42 (21–90) | 70 (30–180) | 90 (30–135) |
| **Smoking status** | | | | | | |
| Never | 460 (62%) | 186 (64%) | 180 (77%) | 35 (78%) | 50 (64%) | 911 (66%) |
| Current or former | 261 (35%) | 95 (33%) | 53 (23%) | 8 (18%) | 25 (32%) | 442 (32%) |
| Missing/unknown | 15 (2%) | 8 (3%) | 2 (1%) | 2 (4%) | 3 (4%) | 30 (2%) |
| **Obesity** | | | | | | |
| No | 421 (57%) | 133 (46%) | 116 (49%) | 32 (71%) | 45 (58%) | 747 (54%) |
| Yes | 308 (42%) | 156 (54%) | 116 (49%) | 13 (29%) | 33 (42%) | 626 (45%) |
| Missing/unknown | 7 (1%) | 0 (0%) | 3 (1%) | 0 (0%) | 0 (0%) | 10 (1%) |
| **Comorbidities**[†] | | | | | | |
| Cardiovascular | 179 (24%) | 60 (21%) | 14 (6%) | 7 (16%) | 11 (14%) | 271 (20%) |
| Pulmonary | 125 (17%) | 65 (22%) | 33 (14%) | <5 (<11%) | 7 (9%) | 234 (17%) |
| Renal disease | 66 (9%) | 31 (11%) | 13 (6%) | <5 (<11%) | <5 (<6%) | 115 (8%) |
| Diabetes mellitus | 127 (17%) | 98 (34%) | 51 (22%) | 10 (22%) | 20 (26%) | 306 (22%) |
| Missing/unknown | 9 (1%) | 1 (<1%) | 5 (2%) | 0 (0%) | 0 (0%) | 15 (1%) |
| **ECOG performance status** | | | | | | |

*Table 1 continued on next page*

*Table 1 continued*

| | NHW | Black | Hispanic | AAPI | Others | All |
|---|---|---|---|---|---|---|
| 0 | 314 (43%) | 130 (45%) | 123 (52%) | 18 (40%) | 32 (41%) | 617 (45%) |
| 1 | 135 (18%) | 72 (25%) | 48 (20%) | 10 (22%) | 16 (21%) | 281 (20%) |
| 2+ | 69 (9%) | 33 (11%) | 15 (6%) | 5 (11%) | 5 (6%) | 127 (9%) |
| Unknown | 218 (30%) | 53 (18%) | 49 (21%) | 12 (27%) | 25 (32%) | 357 (26%) |
| Missing | 0 (0%) | 1 (<1%) | 0 (0%) | 0 (0%) | 0 (0%) | 1 (<1%) |
| **Region** | | | | | | |
| Northeast | 247 (34%) | 101 (35%) | 106 (45%) | 12 (27%) | 26 (33%) | 492 (36%) |
| Midwest | 239 (32%) | 110 (38%) | 23 (10%) | 8 (18%) | 12 (15%) | 392 (28%) |
| South | 116 (16%) | 58 (20%) | 27 (11%) | X* | 14 (18%) | 218 (16%) |
| West | 128 (17%) | 16 (6%) | 77 (33%) | 22 (49%) | 24 (31%) | 267 (19%) |
| Undesignated | 6 (1%) | 4 (1%) | 2 (1%) | 3 (7%)* | 2 (3%) | 14 (1%) |
| **Month/year of COVID-19 diagnosis** | | | | | | |
| Jan-Apr 2020 | 140 (19%) | 74 (26%) | 41 (17%) | 8 (18%) | 20 (26%) | 283 (20%) |
| May-Aug 2020 | 279 (38%) | 141 (49%) | 101 (43%) | 24 (53%) | 30 (38%) | 575 (42%) |
| Sept-Dec 2020 | 197 (27%) | 42 (15%) | 50 (21%) | 5 (11%) | 16 (21%) | 310 (22%) |
| Jan-Jun 2021 | 118 (16%) | 32 (11%) | 41 (17%) | 7 (16%) | 12 (15%) | 210 (15%) |
| Missing/unknown | 2 (<1%) | 0 (0%) | 2 (1%) | 1 (2%) | 0 (0%) | 5 (<1%) |
| **Area of patient residence** | | | | | | |
| Urban | 193 (26%) | 136 (47%) | 124 (53%) | 13 (29%) | 30 (38%) | 496 (36%) |

*Table 1 continued on next page*

*Table 1 continued*

| | NHW | Black | Hispanic | AAPI | Others | All |
|---|---|---|---|---|---|---|
| Suburban | 315 (43%) | 77 (27%) | 65 (28%) | 17 (38%) | 31 (40%) | 505 (37%) |
| Rural | 81 (11%) | 7 (2%) | 9 (4%) | X* | 0 (0%) | 98 (7%) |
| Missing/unknown | 147 (20%) | 69 (24%) | 37 (16%) | 15 (33%)* | 17 (22%) | 284 (21%) |
| **Treatment center characteristics** | | | | | | |
| Academic medical center | 123 (17%) | 102 (35%) | 43 (18%) | 7 (16%) | 11 (14%) | 286 (21%) |
| Community practice | 238 (32%) | 51 (18%) | 44 (19%) | X* | 23 (29%) | 359 (26%) |
| Tertiary care center | 375 (51%) | 136 (47%) | 147 (63%) | 35 (78%) | 44 (56%) | 737 (53%) |
| Missing/unknown | 0 (0%) | 0 (0%) | 1 (<1%) | 3 (7%)* | 0 (0%) | 1 (<1%) |
| **Receptor status** | | | | | | |
| HR+/HER2- | 419 (57%) | 135 (47%) | 102 (43%) | 22 (49%) | 43 (55%) | 721 (52%) |
| HR+/HER2+ | 102 (14%) | 35 (12%) | 43 (18%) | 7 (16%) | 9 (12%) | 196 (14%) |
| HR-/HER2+ | 46 (6%) | 28 (10%) | 32 (14%) | X* | X* | 111 (8%) |
| Triple negative | 57 (8%) | 54 (19%) | 35 (15%) | 5 (11%) | 7 (9%) | 158 (11%) |
| Missing/unknown | 112 (15%) | 37 (13%) | 23 (10%) | 11 (24%) | 19 (24%)* | 197 (14%) |
| **Cancer status** | | | | | | |
| Remission/NED, >5 years | 247 (34%) | 76 (26%) | 23 (10%) | 9 (20%) | 20 (26%) | 375 (27%) |
| Remission/NED, <5 years | 234 (32%) | 100 (35%) | 77 (33%) | 11 (24%) | 26 (33%) | 448 (32%) |
| Active and responding | 68 (9%) | 35 (12%) | 56 (24%) | X* | 11 (14%) | 173 (13%) |
| Active and stable | 91 (12%) | 28 (10%) | 35 (15%) | 10 (22%) | 5 (6%) | 169 (12%) |

*Table 1 continued on next page*

*Table 1 continued*

| | NHW | Black | Hispanic | AAPI | Others | All |
|---|---|---|---|---|---|---|
| Active and progressing | 41 (6%) | 27 (9%) | 20 (9%) | 6 (13%) | X* | 97 (7%) |
| Unknown | 48 (7%) | 19 (7%) | 22 (9%) | 6 (13%)* | 15 (19%)* | 104 (8%) |
| Missing | 7 (1%) | 4 (1%) | 2 (1%) | 3 (7%) | 1 (1%) | 17 (1%) |
| **Timing of anti-cancer therapy** | | | | | | |
| Never/after COVID-19 | 24 (3%) | 10 (3%) | 7 (3%) | X* | 7 (9%) | 50 (4%) |
| 0–4 weeks | 364 (49%) | 135 (47%) | 158 (67%) | 25 (56%) | 39 (50%) | 721 (52%) |
| 1–3 months | 26 (4%) | 20 (7%) | 19 (8%) | 0 (0%) | X* | 69 (5%) |
| >3 months | 303 (41%) | 118 (41%) | 45 (19%) | 18 (40%) | 24 (31%) | 508 (37%) |
| Missing/unknown | 19 (3%) | 6 (2%) | 6 (3%) | 2 (4%)* | 8 (10%)* | 35 (3%) |
| **Modality of active anti-cancer therapy‡, §** | | | | | | |
| None | 333 (45%) | 127 (44%) | 53 (23%) | 20 (44%) | 30 (38%) | 563 (41%) |
| Chemotherapy | 117 (16%) | 68 (24%) | 88 (37%) | 11 (24%) | 14 (18%) | 298 (22%) |
| Targeted therapy | 112 (15%) | 38 (13%) | 59 (25%) | 8 (18%) | 11 (14%) | 228 (16%) |
| Anti-HER2 therapy | 60 (8%) | 17 (6%) | 36 (15%) | <5 (<11%) | <5 (<6%) | 123 (9%) |
| CDK4/6 inhibitor | 33 (4%) | 12 (4%) | 14 (6%) | <5 (<11%) | <5 (<6%) | 65 (5%) |
| Other¶ | 14 (2%) | 5 (2%) | <5 (<2%) | <5 (<11%) | 0 (0%) | 24 (2%) |
| Endocrine therapy | 283 (38%) | 86 (30%) | 91 (39%) | 15 (33%) | 26 (33%) | 501 (36%) |
| Immunotherapy | 12 (2%) | 8 (3%) | <5 (<2%) | <5 (<11%) | <5 (<6%) | 28 (2%) |
| Local (surgery/radiation) | 80 (11%) | 37 (13%) | 41 (17%) | <5 (<11%) | 9 (12%) | 172 (12%) |

*Table 1 continued on next page*

*Table 1 continued*

| | NHW | Black | Hispanic | AAPI | Others | All |
|---|---|---|---|---|---|---|
| Other | 13 (2%) | 3 (1%) | 2 (1%) | 0 (0%) | 0 (0%) | 18 (1%) |
| Missing/unknown | 12 (2%) | 7 (2%) | 5 (2%) | 0 (0%) | 5 (6%) | 29 (2%) |
| **Severity of COVID-19** | | | | | | |
| Mild | 535 (73%) | 177 (61%) | 173 (74%) | 28 (62%) | 50 (64%) | 963 (70%) |
| Moderate | 174 (24%) | 97 (34%) | 56 (24%) | 14 (31%) | 21 (27%) | 362 (26%) |
| Severe | 25 (3%) | 15 (5%) | 6 (3%) | X* | 7 (9%) | 56 (4%) |
| Missing/unknown | 2 (<1%) | 0 (0%) | 0 (0%) | 3 (7%)* | 0 (0%) | 2 (<1%) |

Variable categories with one to five cases are masked by replacing with N < 5 according to CCC19 policy.

*Cells combined to mask N<5 according to CCC19 low count policy.

†Age was truncated at 90.

‡Percentages could sum to >100% because categories are not mutually exclusive.

§Within 3 months of COVID-19 diagnosis.

¶Therapies other than anti-Her2 therapy or CDK4/6 inhibitor.

**Table 2.** Outcomes, clinical complications, and COVID-19 interventions.

| | NHW | Black | Hispanic | AAPI | Other | All |
|---|---|---|---|---|---|---|
| | n** (%) | n** (%) | n** (%) | n** (%) | n** (%) | n** (%) |
| **Outcomes** | | | | | | |
| Total all-cause mortality* | 60 (8) | 38 (13) | 12 (5) | <5(<11) | 9 (12) | 123 (9) |
| 30-day all-cause mortality† | 40 (5) | 29 (10) | 8 (3) | <5 (<11) | 8 (10) | 89 (6) |
| Received mechanical ventilation* | 24 (3) | 26 (9) | 11 (5) | <5 (<11) | <5 (<6) | 69 (5) |
| Admitted to an intensive care unit* | 45 (6) | 31 (11) | 18 (8) | 7 (16) | 10 (13) | 111 (8) |
| Admitted to the hospital* | 245 (33) | 137 (47) | 77 (33) | 20 (44) | 33 (42) | 512 (37) |
| **Clinical complications** | | | | | | |
| Any cardiovascular complication‡ | 82 (11) | 50 (17) | 30 (13) | 6 (13) | 18 (23) | 186 (14) |
| Any pulmonary complication§ | 170 (23) | 88 (31) | 43 (18) | 12 (27) | 23 (30) | 336 (24) |
| Any gastrointestinal complication¶ | 12 (2) | 7 (2) | <5 (<2) | <5 (<11) | <5 (<7) | 26 (2) |
| Acute kidney injury | 41 (6) | 46 (16) | 11 (5) | 5 (11) | 10 (13) | 113 (8) |
| Multisystem organ failure | 10 (1) | 12 (4) | <5 (<2) | <5 (<11) | <5 (<7) | 29 (2) |
| Superimposed infection | 62 (9) | 42 (15) | 14 (6) | 7 (16) | <5 (<7) | 129 (10) |
| Sepsis | 43 (6) | 24 (8) | 15 (6) | 7 (16) | 12 (16) | 101 (7) |
| Any bleeding | 15 (2) | 7 (2) | <5 (<2) | <5 (<11) | <5 (<7) | 29 (2) |
| **Interventions** | | | | | | |
| Remdesivir | 68 (10) | 20 (7) | 15 (7) | 8 (18) | 5 (7) | 116 (9) |
| Hydroxychloroquine | 60 (9) | 41 (15) | 14 (6) | <5 (<11) | 11 (15) | 129 (10) |
| Systemic corticosteroids | 107 (15) | 50 (18) | 31 (14) | 8 (18) | 13 (18) | 209 (16) |
| Other | 112 (16) | 53 (19) | 36 (16) | 11 (25) | 12 (17) | 224 (17) |
| Supplemental oxygen | 173 (24) | 87 (31) | 43 (19) | 14 (31) | 24 (31) | 341 (25) |

Variable categories with one to five cases are masked by replacing with N<5 according to CCC19 policy.

*Included in primary outcome.

†Secondary outcome.

‡Cardiovascular complication includes hypotension, myocardial infarction, other cardiac ischemia, atrial fibrillation, ventricular fibrillation, other cardiac arrhythmia, cardiomyopathy, congestive heart failure, pulmonary embolism (PE), deep vein thrombosis (DVT), stroke, thrombosis NOS complication.

§Pulmonary complication includes respiratory failure, pneumonitis, pneumonia, acute respiratory distress syndrome (ARDS), PE, pleural effusion, empyema.

¶Gastrointestinal complication includes acute hepatic injury, ascites, bowel obstruction, bowel perforation, ileus, peritonitis.

**N based on number of patients with non-missing data.

With regard to baseline severity of COVID-19 at presentation, 39% of Black and 38% of AAPI patients presented with moderate or higher severity of COVID-19 infection compared to 27% in both NHW and Hispanic patients. *Table 2* summarizes the clinical outcomes, complications, and interventions, stratified by race/ethnicity.

## Characteristics of female patients with MBC and COVID-19

Female patients with MBC consisted of 17% of the cohort (N=233), with median age 58 years [IQR 50–68]. Racial/ethnic groups consisted of 46% NHW, 24% Black, 21% Hispanics, 4% AAPI, and 4% Other. Most patients with MBC were never smokers (70%) and non-obese (60%). The predominant tumor biology was HR+/HER2- (42%) followed by HR+/HER2+ (23%). The most common sites of metastases were bone (58%), lung (28%), and liver (26%). A high percentage (87%) had received anti-cancer treatment within 3 months prior to COVID-19 diagnosis and 32% had active and progressing

**Table 3.** Systemic treatments received within 3 months prior to COVID-19 diagnosis.

|  | N (%) |
| --- | --- |
| Total | 679 (100%) |
| Endocrine therapy alone | 336 (49.5) |
| CDK4/6 inhibitor ± endocrine therapy | 63 (9) |
| Other targeted therapy ± endocrine therapy | 10 (1.5) |
| Anti-HER2 therapy ± endocrine therapy | 78 (11.5) |
| Anti-HER2 therapy + chemotherapy | 48 (7) |
| Single agent chemotherapy ± endocrine therapy | 55 (8) |
| Combination chemotherapy ± endocrine therapy | 60 (9) |
| Immunotherapy ± chemotherapy | 19 (3) |
| Other combination therapies | 10 (1.5) |

cancer. The unadjusted total all-cause mortality and hospitalization rate in females with MBC was 19% and 53% respectively. Further details of baseline characteristics and unadjusted rates of COVID-19 outcomes, complications, and interventions are presented in *Appendix 6—table 1* and *Appendix 6—table 2*.

## BC treatment characteristics

758 (55%) out of 1383 female patients with BC received some form of systemic treatment within 3 months prior to COVID-19 diagnosis, and specific drug information was available for 679 (90%) (*Table 3*). Of these 679 patients, the most common systemic therapy was endocrine therapy alone (n=336, 49.5%). This was followed by chemotherapy in 163 (24%) patients who received it either as single agent (n=55, 8%) or combination chemotherapy (n=60, 9%) or combined with anti-HER2 therapy (n=48, 7%). 78 (11.5%) patients received anti-HER2 therapy with or without endocrine therapy, and 63 (9%) patients received CDK4/6 inhibitors with or without endocrine therapy.

## Prognostic factors associated with COVID-19 severity

After adjusting for baseline demographic, clinical, and spatiotemporal factors in multivariable analysis model, factors associated with worse outcomes in females with BC included older age (aOR per decade, 1.48 [95% CI, 1.32–1.67]); Black (aOR, 1.74 [95% CI, 1.24–2.45]), AAPI (aOR, 3.40 [95% CI, 1.70–6.79]), and Other (aOR, 2.97 [95% CI, 1.71–5.17]) racial/ethnic group; cardiovascular (aOR, 2.26 [95% CI, 1.63–3.15]) and pulmonary (aOR, 1.65 [95% CI, 1.20–2.29]) comorbidities; diabetes mellitus (aOR, 2.25 [95% CI, 1.66–3.04]); worse ECOG PS (ECOG PS 1: aOR, 1.74 [95% CI, 1.22–2.48]; ECOG PS ≥2: aOR, 7.78 [95% CI, 4.83–12.5]); and active and progressing cancer status (aOR, 12.5 [95% CI, 6.89–22.6]). Association between Hispanic ethnicity, obesity, pre-existing renal disease, anti-cancer treatment modalities including all forms of systemic therapy and locoregional therapy, month/year, and geographic region of COVID-19 diagnosis and COVID-19 severity did not reach statistical significance (*Table 4*). The e value for the COVID-19 severity OR and CI for each racial group are shown in *Appendix 7—table 1*. This value demonstrates the impact of unknown _residual_ confounding above that adjusted for by including adjustment variables in the multivariable model. For example, an unmeasured confounder would need to be associated with both race and mortality with an OR of at least 1.97 to fully attenuate the observed association for Black females and the OR would need to be at least 1.47 for the null-hypothesized value (1.0) to be included in the CI. Similarly, e value estimates are noted for AAPI and Other groups. The unmeasured confounding for other races based on the e value is larger than most documented associations in the CCC19 cohort (*Grivas et al., 2021*).

## Male patients with BC and COVID-19

Male patients with BC were evaluated separately as part of exploratory analysis. The median age for male BC cohort (N=25) was 67 years [IQR 60–75]. Racial/ethnic composition consisted of NHW (52%) followed by Black (32%) males. Most males with BC were non-smokers (72%) and diabetes mellitus

**Table 4.** Adjusted associations of baseline characteristics with COVID-19 severity outcome.

| | COVID-19 severity |
|---|---|
| | OR (95% CI) |
| Age (per decade) | 1.48 (1.32–1.67) |
| Race (Ref: non-Hispanic White)* | |
| Non-Hispanic Black | 1.74 (1.24–2.45) |
| Hispanic | 1.38 (0.93–2.05) |
| Non-Hispanic AAPI | 3.40 (1.70–6.79) |
| Other | 2.97 (1.71–5.17) |
| Obesity (Ref: No) | 1.20 (0.92–1.57) |
| Cardiovascular comorbidity (Ref: No) | 2.26 (1.63–3.15) |
| Pulmonary comorbidity (Ref: No) | 1.65 (1.20–2.29) |
| Renal disease (Ref: No) | 1.34 (0.86–2.07) |
| Diabetes mellitus (Ref: No) | 2.25 (1.66–3.04) |
| ECOG performance status (Ref: 0) | |
| 1 | 1.74 (1.22–2.48) |
| 2+ | 7.78 (4.83–12.5) |
| Unknown | 2.26 (1.61–3.19) |
| Cancer status (Ref: Remission/NED, >5 years) | |
| Remission or NED, <5 years | 0.91 (0.63–1.33) |
| Active and responding | 1.07 (0.63–1.83) |
| Active and stable | 1.37 (0.82–2.28) |
| Active and progressing | 12.5 (6.89–22.6) |
| Unknown | 1.79 (0.96–3.34) |
| Chemotherapy (Ref: No) | 1.37 (0.91–2.06) |
| Anti-HER2 therapy (Ref: No) | 1.13 (0.67–1.92) |
| CDK 4/6 inhibitor (Ref: No) | 1.21 (0.60–2.42) |
| Other targeted therapies[†] (Ref: No) | 1.78 (0.69–4.59) |
| Endocrine therapy (Ref: No) | 1.00 (0.73–1.37) |
| Locoregional therapy (Ref: No) | 1.36 (0.88–2.10) |
| Never received cancer treatment (Ref: >3 month) | 0.65 (0.28–1.49) |
| Month/year of COVID-19 diagnosis (Ref: Jan-Apr 2020) | |
| May-Aug 2020 | 0.57 (0.41–0.81) |
| Sept-Dec 2020 | 0.45 (0.30–0.68) |
| Jan-Jun 2021 | 0.57 (0.36–0.89) |
| Region (Ref: Northeast) | |
| Midwest | 0.76 (0.54–1.05) |
| South | 0.76 (0.51–1.13) |
| West | 0.43 (0.29–0.65) |

*Odds ratios greater than 1 indicate higher odds of composite outcome. The p value for evaluating the null hypothesis of equality in odds ratios across race (4 degrees of freedom) was <0.001.

[†]Therapies other than CDK4/6 inhibitor or anti-HER2 therapy. All variance inflation factors are <1.8 for the model.

was the predominant comorbidity (44%). The hospitalization rate was 60% and all-cause mortality was 20%. Additional clinical characteristics, complications, interventions, and unadjusted outcomes among males with BC in the CCC19 registry are provided in *Appendix 8—table 1* and *Appendix 8— table 2*.

## Discussion

In this large, multi-institutional and racially diverse cohort of females with BC and COVID-19 from CCC19 registry, we assessed the clinical impact of COVID-19. The all-cause mortality from COVID-19 was 9% and hospitalization rate was 37%, which is numerically lower than in the entire CCC19 cohort at 14% and 58%, and other previously reported studies of COVID-19 in patients with cancer (*Grivas et al., 2021*; *Garassino et al., 2020*; *Lee et al., 2020*; *Wang et al., 2021*; *de Azambuja et al., 2020*; *Albiges et al., 2020*; *Zhang et al., 2021*). These differences in outcomes could indicate differences in the immunocompromised status of patients due to intensity of therapy regimens, complex comorbidities, or concomitant medications, which may affect outcomes. Females with BC, however, form a heterogenous group, and the rates of outcomes varied widely with their disease status; patients with active and progressing cancer had the highest total all-cause mortality (38%) and hospitalization rates (72%).

We observed older age, pre-existing cardiovascular and pulmonary comorbidities, diabetes mellitus, worse ECOG PS, and active and progressing cancer status were associated with adverse COVID-19 outcomes in females with BC. Prior studies have reported similar factors to be associated with adverse COVID-19 outcomes in patients with all cancer types. The majority of these studies have reported older age to be an important prognostic factor for adverse outcomes from COVID-19, including mortality, which is consistent with data presented here (*Grivas et al., 2021*; *Sharafeldin et al., 2021*; *Lièvre et al., 2020*; *Zhang et al., 2021*; *Chavez-MacGregor et al., 2022*). Non-cancer comorbidities, contributing to poor COVID-19 outcomes, as noted in our study, have also been a consistent finding in patients with and without a cancer diagnosis (*Grivas et al., 2021*; *Sharafeldin et al., 2021*; *Lièvre et al., 2020*; *Chavez-MacGregor et al., 2022*; *CDC, 2020c*). Similarly, poor ECOG PS in cancer patients has been noted to be an important factor associated with worse COVID-19 severity, including our study (*Grivas et al., 2021*; *Albiges et al., 2020*; *Lièvre et al., 2020*). While obesity was reported in some cancer studies to have a negative impact on COVID-19 (*Grivas et al., 2021*; *Chavez-MacGregor et al., 2022*), our study did not identify this association. In this cohort of females with BC, all forms of anti-cancer therapy were thoroughly evaluated and none of the systemic therapies including chemotherapy, endocrine therapy, and targeted therapy (anti-HER2, CDK4/6 inhibitors, other non-HER2 or non-CDK4/6 inhibitors), or locoregional therapy (surgery and radiation) received within 3 months of COVID-19 diagnosis was significantly associated with adverse COVID-19 outcomes. Our finding suggests that systemic therapy for females with BC may not add excess COVID-19 risk. Multiple large cohort studies and meta-analysis of patients with cancer diagnosed with COVID-19 similarly did not identify active anti-cancer therapy, specifically chemotherapy, as a factor associated with adverse COVID-19 outcomes, which is consistent with our results (*Garassino et al., 2020*; *Lee et al., 2020*; *Albiges et al., 2020*; *Zhang et al., 2021*; *Liu et al., 2021*; *Jee et al., 2020*). However, in contrast, some studies of patients with other cancers have shown a negative impact of chemotherapy (*Grivas et al., 2021*; *Sharafeldin et al., 2021*; *Lièvre et al., 2020*; *Chavez-MacGregor et al., 2022*) and immunotherapy use (*Chavez-MacGregor et al., 2022*). These findings have important clinical implications while counselling and providing patient care during the pandemic.

We also report important findings related to the impact of racial/ethnic inequities in females with BC and COVID-19, which adds to the growing body of literature on COVID-19-related racial/ethnic disparities. In our study, Black females with BC had significantly worse COVID-19 outcomes compared to NHW females. Multiple studies have similarly reported Black patients in US with and without cancer diagnosis having significantly worse COVID-19 outcomes (*Grivas et al., 2021*; *Wang et al., 2021*; *CDC, 2020b*); however, our study is the first to show such racial/ethnic disparities in COVID-19 outcomes in females with BC. There was no statistically significant association of worse outcomes for Hispanic females compared to NHW females. This is different in comparison to our overall CCC19 cohort (*Grivas et al., 2021*), and may be explained by younger age and lower rates of comorbid conditions in Hispanic females compared to NHW females. We also found females belonging to AAPI, and Other racial/ethnic group to have worse COVID-19 outcomes. Notably, females belonging

to Black, AAPI, and Other racial/ethnic groups presented with higher rates of moderate or severe symptoms of COVID-19 at baseline, which likely contributed to their worse outcomes. This in turn is possibly related to barriers to health care access, and other socio-cultural reasons for delay in seeking early medical care. Future studies including social determinants of health, access to health care, and lifestyle behaviors, among others, are warranted to identify barriers contributing to worse clinical presentation in racial/ethnic minority groups, and eventually impacting future health policies.

In summary, this is one of the largest cohort studies to evaluate the clinical impact of COVID-19 on females with BC. Strengths of our study include standardized data collection on the most common cancer in females in the US and large sample size to evaluate the effect of major clinical and demographic factors. The study had representative population by race and ethnicity from geographically diverse areas and variable time/period of COVID-19 diagnosis. In addition, our study has detailed manually collected information on both cancer status and treatment modalities which contrasts with other studies that have utilized either of these variables as surrogate. Limitations of this study include the retrospective nature of data and inherent potential for confounding because of its observational nature. It's possible that ascertainment bias could have led to some of the high values observed in specific groups such as females with MBC and those with active and progressing cancer. Additional information on drivers for inequity such as socio-economic status, occupation, income, residence, education, and insurance status may have provided added insights on the root causes for disparities; however, unavailability of these factors does not nullify our current findings of existing racial disparities in COVID-19 outcomes in females with BC. Vaccination status was not part of this study as vaccines were not available during the predominant time frame for this cohort. Data presented here including the risk of hospitalization and death applies to the specific COVID-19 variants prevalent during the study period. Despite these limitations, the study reports important sociodemographic and clinical factors that aid in identifying females with BC who are at increased risk for severe COVID-19 outcomes. Given the largely unknown long-term impact of this novel virus, systematic examination of the post-acute sequelae of COVID-19 in patients with breast and other cancer subtypes is warranted.

Our study addresses an important knowledge gap in patients with BC diagnosed with COVID-19 using the CCC19 registry. In addition to clinical and demographic factors associated with adverse COVID-19 outcomes, racial/ethnic disparities reported here significantly contribute to the growing literature. At this stage, it is irrefutable that one of the principal far-reaching messages the pandemic has conveyed is that any such major stressors on the health care system increases risk of detrimental outcomes to the most vulnerable patient population, including the underrepresented and the underserved. These are important considerations for future resource allocation strategies and policy interventions. We also report an important finding that cancers that are active and progressing are associated with severe COVID-19 outcomes. During the ongoing pandemic, this has significant implications for shared decision-making between patients and physicians.

## Acknowledgements

We thank all members of the CCC19 steering committee: Toni K Choueiri, Narjust Duma, Dimitrios Farmakiotis, Petros Grivas, Gilberto de Lima Lopes Jr, Corrie A Painter, Solange Peters, Brian I Rini, Dimpy P Shah, Michael A Thompson, and Jeremy L Warner, for their invaluable guidance of the CCC19.

## Additional information

### Funding

| Funder | Grant reference number | Author |
| --- | --- | --- |
| National Cancer Institute | P30 CA068485 | Tianyi Sun<br>Sanjay Mishra<br>Benjamin French<br>Jeremy L Warner |
| National Cancer Institute | P30-CA046592 | Christopher R Friese |
| National Cancer Institute | P30 CA023100 | Rana R McKay |

| Funder | Grant reference number | Author |
|---|---|---|
| National Cancer Institute | P30-CA054174 | Pankil K Shah<br>Dimpy P Shah |
| American Cancer Society | MRSG-16-152-01 -CCE | Dimpy P Shah |
| National Center for AdvancingTranslational Sciences, National Institute of Health, | KL2 TR002646 | Pankil K Shah |

The funders had no role in study design, data collection and interpretation, or the decision to submit the work for publication.

## Author contributions

Gayathri Nagaraj, Shaveta Vinayak, Ali Raza Khaki, Maryam B Lustberg, Melissa K Accordino, Conceptualization, Resources, Data curation, Supervision, Investigation, Visualization, Methodology, Writing – original draft, Project administration, Writing – review and editing; Tianyi Sun, Data curation, Software, Formal analysis, Validation, Visualization, Methodology, Writing – original draft, Writing – review and editing; Nicole M Kuderer, Conceptualization, Writing – review and editing; David M Aboulafia, Jared D Acoba, Joy Awosika, Ziad Bakouny, Nicole B Balmaceda, Ting Bao, Babar Bashir, Stephanie Berg, Mehmet A Bilen, Poorva Bindal, Sibel Blau, Brianne E Bodin, Hala T Borno, Cecilia Castellano, Horyun Choi, John Deeken, Aakash Desai, Natasha Edwin, Lawrence E Feldman, Daniel B Flora, Matthew D Galsky, Cyndi J Gonzalez, Petros Grivas, Shilpa Gupta, Marcy Haynam, Hannah Heilman, Dawn L Hershman, Clara Hwang, Chinmay Jani, Sachin R Jhawar, Monika Joshi, Virginia Kaklamani, Elizabeth J Klein, Natalie Knox, Vadim S Koshkin, Amit A Kulkarni, Daniel H Kwon, Chris Labaki, Philip E Lammers, Kate I Lathrop, Mark A Lewis, Xuanyi Li, Gilbert de Lima Lopes, Gary H Lyman, Della F Makower, Abdul-Hai Mansoor, Merry-Jennifer Markham, Sandeep H Mashru, Ian Messing, Vasil Mico, Rajani Nadkarni, Swathi Namburi, Ryan H Nguyen, Taylor Kristian Nonato, Tracey Lynn O'Connor, Orestis A Panagiotou, Kyu Park, Jaymin M Patel, Kanishka GopikaBimal Patel, Jeffrey Peppercorn, Hyma Polimera, Matthew Puc, Yuan James Rao, Pedram Razavi, Sonya A Reid, Jonathan W Riess, Donna R Rivera, Mark Robson, Suzanne J Rose, Atlantis D Russ, Lidia Schapira, M Kelly Shanahan, Lauren C Shapiro, Melissa Smits, Daniel G Stover, Mitrianna Streckfuss, Lisa Tachiki, Michael A Thompson, Sara M Tolaney, Lisa B Weissmann, Grace Wilson, Michael T Wotman, Elizabeth M Wulff-Burchfield, Data curation, Writing – review and editing; Christopher R Friese, Rana R McKay, Pankil K Shah, Data curation, Funding acquisition, Writing – review and editing; Sanjay Mishra, Project administration, Writing – review and editing; Benjamin French, Conceptualization, Data curation, Software, Formal analysis, Validation, Visualization, Methodology, Writing – original draft, Writing – review and editing; Jeremy L Warner, Conceptualization, Resources, Data curation, Supervision, Funding acquisition, Methodology, Writing – review and editing; Dimpy P Shah, Conceptualization, Resources, Data curation, Supervision, Funding acquisition, Investigation, Visualization, Methodology, Writing – original draft, Project administration, Writing – review and editing

## Author ORCIDs

Gayathri Nagaraj (iD) https://orcid.org/0000-0002-7586-6920
Tianyi Sun (iD) http://orcid.org/0009-0005-2695-0885
Clara Hwang (iD) http://orcid.org/0000-0002-0998-323X
Matthew Puc (iD) http://orcid.org/0000-0002-9281-2262
Sanjay Mishra (iD) http://orcid.org/0000-0002-7775-9600
Melissa K Accordino (iD) http://orcid.org/0000-0002-4156-7279

## Ethics

Human subjects: This study was exempt from institutional review board (IRB) review (VUMC IRB#200467) and was approved by IRBs at participating sites per institutional policy. CCC19 registry is registered on ClinicalTrials.gov, NCT04354701.

## Decision letter and Author response

Decision letter https://doi.org/10.7554/eLife.82618.sa1
Author response https://doi.org/10.7554/eLife.82618.sa2

## Additional files

### Supplementary files
- MDAR checklist

### Data availability
All datasets (with restriction of time variables to protect patient confidentiality) and code associated with the article are available at: https://doi.org/10.5061/dryad.1g1jwsv10.

The following dataset was generated:

| Author(s) | Year | Dataset title | Dataset URL | Database and Identifier |
| --- | --- | --- | --- | --- |
| Nagaraj G, Khaki A, Shah DP | 2023 | Covid-19 and Cancer Consortium (CCC19) breast cancer and racial disparities outcomes study | https://doi.org/10.5061/dryad.1g1jwsv10 | Dryad Digital Repository, 10.5061/dryad.1g1jwsv10 |

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

## Appendix 1

### CCC-19 quality scores

The CCC-19 uses a quality scoring system to determine the suitability of records for inclusion in analyses. A score greater than 5 was considered insufficient for inclusion in the analysis presented. Scores are tabulated as follows:

| |
|---|
| **Minor problems (+1 points per problem)** |
| ADT missing/unknown (prostate cancers only) |
| Biomarkers missing/unknown (breast cancers only) |
| ICU admission missing/unknown |
| Hospitalization missing/unknown |
| Mechanical ventilation missing/unknown |
| $O_2$ ever needed missing/unknown |
| Days to death missing/unknown |
| Cancer status unknown |
| ECOG PS unknown |
| Missing cancer drug names for patients on systemic anti-cancer treatment |
| Missing or unknown categorical lab values if labs were drawn |
| Moderate problems (+3 points per problem) |
| Cancer status missing |
| ECOG PS missing |
| Death status missing/unknown |
| Baseline COVID-19 severity missing/unknown |
| Should have 30-day follow-up but doesn't |
| Major problems (+5 points per problem) |
| High levels of missingness |
| High levels of unknowns |

## Appendix 2

### Breast cancer disparities statistical analysis plan

Approved Project Title: Racial and Ethnic Disparities among Patients with Breast Cancer and COVID-19 in CCC19 Cohort

Project Team Leads: Gayathri Nagaraj, Melissa Accordino, Maryam Lustberg, Dimpy Shah

Name of the investigator completing this survey: Gayathri Nagaraj and Melissa Accordino

Proposed milestone deadline for this manuscript:

- Abstract submission for ASCO 2021, deadline February 17 completed.
- ASCO abstract accepted for oral presentation. Deadlines for prelim slide upload May 7, and final deadline for uploading slides May 14.
- Manuscript preparation simultaneously, deadline and journal TBD

Do you have local statistical support: No

Name and emails of (at most) two additional project team members who would like to be part of the analysis team for the project:

Melissa Accordino, Email: mkg2134@cumc.columbia.edu
Maryam Lustberg, Email: Maryam.Lustberg@osumc.edu
Dimpy Shah, Email: shahdp@uthscsa.edu

Initial draft of the Statistical Analysis Plan (SAP), following STROBE guidelines, for our review and input. Please complete sections 1 and 3–11 (and 12 if you have local statistical support)

1 (a) Manuscript Title: Racial and Ethnic Disparities among Patients with Breast Cancer and COVID-19 in CCC19 Cohort

1 (b) Provide in the abstract an informative and balanced summary of what was done and what will be found.

Racial and ethnic minority subgroups are at a disproportionately increased risk of contracting COVID-19 or experiencing severe illness regardless of age. Racial and ethnic disparities also affect breast cancer incidence and mortality. The impact of COVID-19 on patients with breast cancer is largely unknown but is currently under investigation. Outcomes of COVID-19 specifically in racial and ethnic minority patients with active or prior history of breast cancer are currently unknown.

## 3. Objectives

### State-specific objectives, including any prespecified hypotheses

The overarching goal of this study is to evaluate the racial and ethnic disparities related to COVID-19 outcomes, in patients with active or previous history of breast cancer. To evaluate this, the following specific aims are proposed:

- Specific Aim 1: To compare the distribution of major clinical, sociodemographic, and breast cancer risk factors among racial and ethnic subgroups of women with active or previous history of single primary invasive breast cancer diagnosed with COVID-19.

  We hypothesize that racial and ethnic minority women with breast cancer are more likely to have active comorbid conditions, such as diabetes mellitus, obesity, smoking history, and a baseline lower performance status compared to NHW women with active or previous history of breast cancer diagnosed with COVID-19. Other variables of interest are age, month/year of COVID-19 diagnosis, area of patient residence, geographic region, insurance type, treatment center characteristics, receipt of anti-COVID-19 treatment along with tumor characteristics including breast cancer biologic subtype, cancer status, treatment intent, timing of anti-cancer treatment, and modality of anti-cancer treatment.

- Specific Aim 2: To compare COVID-19 clinical outcomes on a five-level ordinal scale based on patient's most severe reported outcomes: no complications (uncomplicated); hospital admission, ICU admission, mechanical ventilation; or death from any cause in racial and ethnic minority subgroups of women with previous or active history of breast cancer compared to NHW adjusted for baseline characteristics. We also plan to evaluate the death within 30 days of COVID-19 diagnosis among racial and ethnic subgroups of women with previous or active history of breast cancer compared to NHW adjusted for baseline characteristics.

We *hypothesize* that there will be higher rates of severe COVID-19-related outcomes in the racial and ethnic minority subgroups compared to NHW patients with active or previous history of breast cancer.

- Exploratory aims:
  1. To evaluate the frequency of hospitalization, supplemental oxygen use, ICU admission, and use of mechanical ventilation in the various racial ethnic groups.
  2. To describe the distribution of major clinical, sociodemographic, breast cancer risk factors and outcomes in men with active or previous history of breast cancer diagnosed with COVID-19.
  3. Assess the rate of major clinical complications such as cardiovascular, pulmonary, gastrointestinal, superimposed infection, vascular thrombosis, and others among various racial and ethnic groups of women with active or previous history of breast cancer.

## 4. Study design
## Present key elements of study design early in the paper

This is a retrospective cohort study using de-identified data from the CCC19 database which is a centralized multi-institution registry of patients with current or past history of cancer diagnosed with COVID-19. Study data are collected and managed using REDCap software hosted at Vanderbilt University Medical Center.

## 5. Setting
## Describe the setting, locations, and relevant dates, including periods of recruitment, exposure, follow-up, and data collection

The CCC19 international registry consists of de-identified data on adult patients (18 years and older) with a current or past history of hematologic malignancy or invasive solid tumor who either have laboratory-confirmed SARS-CoV-2 infection or presumptive diagnosis of COVID-19. The CCC19 registry includes patients with either active cancer or a history of cancer and contains variables related to patient demographics, cancer history, and COVID-19 clinical course including receipt of COVID-19-related therapeutics along with follow-up data. The member institutions of the consortium report data through the online REDCap data collection survey developed by CCC19. Data collection period is ongoing, for the purpose of this analysis, the data collected from March 17, 2020, to February 9, 2021, will be used.

## 6. Participants
## (a) Give the eligibility criteria, and the sources and methods of selection of participants. Describe methods of follow-up

Patients with active or previous history of invasive breast cancer with evaluable self-reported race/ethnicity data, and with laboratory-confirmed COVID-19 will be our study population. Primary analysis will be restricted to women with active or previous history of breast cancer. Descriptive data on men with active or previous history of breast cancer will be provided separately as part of the exploratory analysis given the small numbers. We will restrict our analysis to patients diagnosed in the US since the racial and ethnic disparities of interest have been previously described in the US. We will also exclude patients who have multiple malignancies including a history of bilateral breast cancer with the exception of contralateral DCIS only. Further, patients who are not evaluable for the primary ordinal outcome or with a data quality score >4 will be excluded. For this analysis, the unknown/not reported category of race and ethnicity will be excluded.

## (b) For matched studies, give matching criteria and number of exposed and unexposed

Not applicable as the CCC19 registry does not carry data for cancer patients who are not exposed to COVID-19.

## 7. Variables (clearly define all variables)

## Outcomes

- Primary: COVID-19 severity outcome defined on a five-level ordinal scale based on patient's most severe reported outcomes: no complications (uncomplicated); hospital admission; ICU admission, mechanical ventilation; or death from any cause.
- Secondary: 30-day all-cause mortality
- Exploratory/descriptive:
  - Rates of hospitalization; oxygen requirements; ICU admission; mechanical ventilation.
  - Major clinical complications (cardiovascular, pulmonary, gastrointestinal, AKI, MOF, superimposed infection, sepsis, any bleeding, DIC, thrombosis).
  - Descriptive statistics for men with breast cancer diagnosed with COVID-19.

## Exposures

## Predictors

1. Self-reported race
2. Self-reported ethnicity

## **Potential confounders**

Higher priority

1. Age in years
2. Obesity (obese, not obese)
3. Comorbidities (pulmonary, cardiovascular, renal, diabetes mellitus)
4. ECOG PS (0, 1, ≥2, unknown)
5. Receptor status (HR positive, HER2 positive, dual positive, triple negative)
6. Cancer status (remission <5 years, remission >5 years, active stable, active responding, active progressing, unknown)
7. Timing of anti-cancer treatment (never treated, 0–4 weeks, 1–3 months, >3 months)
8. Modality of recent anti-cancer treatment (none, cytotoxic chemotherapy, targeted therapy, endocrine therapy, immunotherapy, locoregional therapy, other)
9. Period of COVID-19 diagnosis (Jan-April 2020, May-August 2020, Sep-Nov 2020, Dec 2020-Feb 2021)

Lower priority
10. Smoking (ever, never)
11. US region of patient residence (NE, MW, South, West)
12. Area of patient residence (urban, suburban, rural)
13. Insurance status (not insured, private insurance, Medicaid/Medicare, other government, missing/unknown)
14. Treatment center characteristics academic (university, tertiary, and NCI designated comprehensive cancer centers), community (practice and hospital), other.

## Effect modifiers

None.

Diagnostic criteria (if applicable).

## 8. Data sources/measurement

For each variable of interest, give sources of data and details of methods of assessment (measurement). Describe comparability of assessment methods if there is more than one group.

## 9. Bias

Describe any efforts to address potential sources of bias.

Multivariable regression models will be used to adjust for known confounding variables.

## 10. Study size
Explain how the study size was arrived at

Study size is based on the number of breast cancer cases reported in the registry at the time of analysis. Breast cancer is the single largest solid tumor cohort within the CCC19 registry accounting for roughly 21% of cases. The numbers are expected to rise given the steep accrual rate.

## 11. Quantitative variables
Explain how quantitative variables will be handled in the analyses. If applicable, describe which groupings will be chosen and why.

## 12. Statistical methods
(a) Describe all statistical methods, including those to be used to control for confounding

### Primary analysis among women

Standard descriptive statistics will summarize major clinical, demographic, and breast cancer prognostic factors; clinical complications during hospitalization; and rates of 30-day mortality, hospitalization, oxygen requirement, ICU admission, and mechanical ventilation among racial and ethnic subgroups. Multivariable ordinal and binary logistic regression models will estimate differences in adjusted odds of COVID-19 severity and 30-day mortality, respectively, between racial and ethnic subgroups. Because the ordinal outcome is assessed over patient's total follow-up period, the model will include an offset for (log) follow-up time. Adjustment covariates will be selected first from the 'higher priority' confounders listed above, followed by those listed as 'lower priority'. Coefficients and standard errors from models with different levels of adjustment, variance inflation factors, and clinical judgement will be used to assess model stability.

### Descriptive analysis among men

We will calculate standard descriptive statistics for major clinical, demographic, and breast cancer prognostic factors and clinical complications during hospitalization; rates of 30-day mortality, hospitalization, oxygen requirement, ICU admission, and mechanical ventilation among men with active or previous history of breast cancer.

(b) Describe any methods that will be used to examine subgroups and interactions

None included.

(c) Explain how missing data will be addressed

Multiple imputation will be used to impute missing and unknown data for all variables included in the analysis, with some exceptions: unknown ECOG performance score and unknown cancer status will not be imputed and treated as a separate category in analyses. Imputation will be performed on the largest dataset possible (i.e., after removing test cases and other manual exclusions, but before applying specific exclusion criteria). At least 10 imputed datasets will be used.

(d) If applicable, explain how loss to follow-up will be addressed

All observed outcomes will be used with models adjusted for duration of follow-up.

(e) Describe any sensitivity analyses

None.

## Appendix 3

## CCC19 approved project variables

**Appendix 3—table 1.** Primary outcome.

| Outcome description | Outcome variable name | Outcome values |
|---|---|---|
| Custom ordinal outcome with death at any time | der_ordinal_v1a | 0=not hospitalized; 1=hospitalized; 2=ICU; 3=mechanical ventilation; 4=death at any time |
| Follow-up in days, with some estimation for intervals | der_days_fu | Integer (days) |

**Appendix 3—table 2.** Secondary outcome.

| Outcome description | Outcome variable name | Outcome values | Additional Details |
|---|---|---|---|
| Derived dead/alive variable | der_deadbinary | 0=No; 1=Yes; 99=Unknown | |
| Derived variable indicating whether patient has died within 30 days of COVID-19 diagnosis (default = No) | der_dead30 | 0=No; 1=Yes; 99=Unknown | |
| Derived variable indicating whether patients required mechanical ventilation | der_mv | 0=No; 1=Yes; 99=Unknown | |
| Derived variable indicating time in ICU | der_ICU | 0=No; 1=Yes; 99=Unknown | |
| Derived hospitalized/not hospitalized variable | der_hosp | 0=No; 1=Yes; 99=Unknown | |
| Derived cardiovascular complication variable (see additional details) | der_CV_event_v2 (der_any_CV is the variable name in R script) | 0=No; 1=Yes; 99=Unknown | Derived with the following derived variables: der_hotn_comp, der_MI_comp, der_card_isch_comp, der_AFib_comp, der_VF_comp, der_arry_oth_comp, der_CMY_comp, der_CHF_comp, der_PE_comp, der_DVT_comp, der_stroke_comp, der_thrombosis_NOS_comp Coded as 1 if any of these variables is 1; coded as 0 if all these variables are 0; coded as 99 if any of variables is 99 and der_CV_event_v2 is missing; otherwise, NA For all listed variable here: 0=No, 1=Yes, 99=Unknown |
| Derived pulmonary complication variable (see additional details) | der_pulm_event (der_any_Pulm is the variable name in R script) | 0=No; 1=Yes; 99=Unknown | Derived with the following derived variables: der_resp_failure_comp, der_pneumonitis_comp, der_pneumonia_comp, der_ARDS_comp, der_PE_comp, der_pleural_eff_comp, der_empyema_comp Coded as 1 if any of these variables is 1; coded as 0 if all these variables are 0; coded as 99 if any of variables is 99 and der_pulm_event is missing; otherwise, NA For all listed variable here: 0=No, 1=Yes, 99=Unknown |

*Appendix 3—table 2 Continued on next page*

*Appendix 3—table 2 Continued*

| Outcome description | Outcome variable name | Outcome values | Additional Details |
|---|---|---|---|
| Derived gastrointestinal complication variable (see additional details) | der_GI_event (der_any_Gast is the variable name in R script) | 0=No; 1=Yes; 99=Unknown | Derived with the following derived variables: der_AHI_comp, der_ascites_comp, der_BO_comp, der_bowelPerf_comp, der_ileus_comp, der_peritonitis_comp. Coded as 1 if any of these variables is 1; coded as 0 if all these variables are 0; coded as 99 if any of variables is 99 and der_GI_event is missing; otherwise, NA. For all listed variable here: 0=No, 1=Yes, 99=Unknown |
| Acute kidney injury (checkbox only) | der_AKI_comp | 0=No; 1=Yes; 99=Unknown | |
| Multisystem organ failure | der_MOF_comp | 0=No; 1=Yes; 99=Unknown | |
| Any co-infection within ±2 weeks of COVID-19 dx | der_coinfection_any | 0=No; 1=Yes; 99=Unknown | |
| Sepsis | der_sepsis_comp | 0=No; 1=Yes; 99=Unknown | |
| Bleeding | der_bleeding_comp | 0=No; 1=Yes; 99=Unknown | |
| DIC (without modifier of definite/probable/possible) | der_DIC_comp | 0=No; 1=Yes; 99=Unknown | |
| Remdesivir as treatment for COVID-19 ever | der_rem | 0=No; 1=Yes; 99=Unknown | |
| Hydroxychloroquine as COVID-19 treatment ever | der_hcq | 0=No; 1=Yes; 99=Unknown | |
| Steroids as COVID-19 treatment ever | der_steroids_c19 | 0=No; 1=Yes; 99=Unknown | |
| COVID-19 treatments other than HCQ, steroids, remdesivir | der_other_tx_c19_v2 | 0=No; 1=Yes; 99=Unknown | |
| Indicates whether patient has ever had supplemental o2 | der_o2_ever | 0=No; 1=Yes; 99=Unknown | |

**Appendix 3—table 3.** Covariate description.

| Covariate description | Variable name | Covariate values | Additional details |
|---|---|---|---|
| Race/ethnicity including Asian | *der_race_v2* | Hispanic; Non-Hispanic AAPI; Non-Hispanic Black; Non-Hispanic White; Other | |
| Age with imputation for categoricals | *der_age_trunc* | Years (continuous 18–89; patients noted to be greater than 89 are set to be age = 90) | |
| Insurance type | der_insurance | Medicaid alone; Medicare alone; Medicare/Medicaid ± other; Other government ± other; Private ± other; Uninsured; Unknown | |

*Appendix 3—table 3 Continued on next page*

*Appendix 3—table 3 Continued*

| Covariate description | Variable name | Covariate values | Additional details |
|---|---|---|---|
| Derived variable for smoking status collapsing the current/former smoker variables | der_smoking2 | Never; Current or Former; Unknown | |
| Binary obesity (BMI ≥ 30 or checkbox checked) indicator | der_obesity | 0=No; 1=Yes; 99=Unknown | |
| Cardiovascular comorbidity (CAD, CHF, Afib, arrhythmia NOS, PVD, CVA, cardiac disease NOS) | der_card | 0=No; 1=Yes; 99=Unknown | |
| Derived variable indicating whether patient has pulmonary comorbidities | der_pulm | 0=No; 1=Yes; 99=Unknown | |
| Renal comorbidities | der_renal | 0=No; 1=Yes; 99=Unknown | |
| Derived variable indicating whether patient has diabetes mellitus | der_dm2 | 0=No; 1=Yes; 99=Unknown | |
| Performance status | der_ecogcat2 | ECOG 0, 1, or 2+ | |
| Breast biomarkers combined variable | der_breast_biomarkers | 1=ER + ; 2=ER + /HER2+; 3=HER2+; 4=triple negative; 99=Unknown | |
| Derived variable indicating cancer status (splits remission/NED by cancer timing) | der_cancer_status_v4 | 0 - Remission/NED, remote; 1 - Remission/NED, recent; 2 - Active, responding; 3 - Active, stable; 4 - Active, progressing; 99 - Unknown | |
| Timing of cancer treatment relative to COVID-19, collapsed | der_cancer_tx_timing_v2 | 0=more than 3 months; 1=0–4 weeks; 2=1–3 months (*); 88=never or after COVID-19 diagnosis; 99=unknown | |
| No cancer treatment in the 3 months prior to COVID-19 | der_cancertr_none | 0=No; 1=Yes; 99=Unknown | Derived with the following covariates: der_any_cyto, der_any_targeted, der_any_endo, der_any_immuno, der_any_local, der_any_other Coded as 1 if all these variables are 0; coded as 0 if any of these variables is 1; coded as 99 if any of these variables is 99; otherwise, NA |
| Any cytotoxic cancer treatment in the 3 months prior to COVID-19 | der_any_cyto | 0=No; 1=Yes; 99=Unknown | |
| Any targeted therapy in the 3 months prior to COVID-19 | der_any_targeted | 0=No; 1=Yes; 99=Unknown | |

*Appendix 3—table 3 Continued on next page*

*Appendix 3—table 3 Continued*

| Covariate description | Variable name | Covariate values | Additional details |
|---|---|---|---|
| Any targeted therapy includes an anti-HER2 therapy in the 3 months prior to COVID-19 | der_her2_3 m | 0=No; 1=Yes | Derived with der_her2, der_any_targeted.<br>Coded as 1 if der_any_targeted is 1 and der_her2 is 1<br>Coded as 0 if: a. der_any_targeted is 1 and der_her2 is 0<br>der_any_targeted is 1<br>Otherwise, NA der_her2:<br>0=No; 1=Yes |
| Any targeted therapy includes a CDK4/6 inhibitor therapy in the 3 months prior to COVID-19 | der_cdk46i_3 m | 0=No; 1=Yes | Derived with der_cdk46i, der_any_targeted.<br>Coded as 1 if der_any_targeted is 1 and der_cdk46i is 1<br>Coded as 0 if: a. der_any_targeted is 1 and der_cdk46i is 0<br>der_any_targeted is 1<br>Otherwise, NA der_cdk46i:<br>0=No; 1=Yes |
| Any other targeted therapy (not anti-HER2/CDK4/6 inhibitor) in the 3 months prior to COVID-19 | der_other_3 m | 0=No; 1=Yes | Derived with der_targeted_not_her2_cdk46i, der_any_targeted.<br>Coded as 1 if der_any_targeted is 1 and der_targeted_not_her2_cdk46i is 1<br>Coded as 0 if: a. der_any_targeted is 1 and der_targeted_not_her2_cdk46i is 0<br>der_any_targeted is 1<br>Otherwise, NA der_targeted_not_her2_cdk46i:<br>0=No; 1=Yes |
| Any endocrine therapy in the 3 months prior to COVID-19 | der_any_endo | 0=No; 1=Yes; 99=Unknown | |
| Any immunotherapy in the 3 months prior to COVID-19 | der_any_immuno | 0=No; 1=Yes; 99=Unknown | |
| Any local therapy (surgery or RT) within 3 months | der_any_local | 0=No; 1=Yes; 99=Unknown | |
| Any other cancer therapy in the 3 months prior to COVID-19 | der_any_other | 0=No; 1=Yes; 99=Unknown | |
| Region of patient residence with ex-US collapsed | der_region_v2 | Non-US; Other; Undesignated US; US Midwest; US Northeast; US South; US West | |
| Trimester and year of diagnosis, using the most recent side of the interval as anchor | der_tri_rt_dx | T1 2020; T2 2020; T3 2020; T1 2021 | |
| What type of area does the patient primarily reside in? | urban_rural[1] | 1, Urban (city) \| 2, Suburban (town, suburbs) \| 3, Rural (country) \| 88, Other \| 99, Unknown | |
| The type of health care center providing the patient's data | der_site_type | AMC = academic medical center; CP = community practice; TCC = tertiary care center | |

*Appendix 3—table 3 Continued on next page*

*Appendix 3—table 3 Continued*

| Covariate description | Variable name | Covariate values | Additional details |
|---|---|---|---|
| Initial severity and course of illness | severity_of_covid_19_v2[1] | 1, Mild (no hospitalization required) \| 2, Moderate (hospitalization indicated) \| 3, Severe (ICU admission indicated) \| 99, Unknown | |
| Derived treatment intent | der_tr_intent | Unknown Treatment; Not on Treatment; Palliative; Curative; Missing<br>Unknown Treatment and Missing were collapsed for analysis | Derived with der_anytx and treatment_intent:<br>Coded as 'Unknown Treatment' if der_anytx is NA or 99;<br>Coded as 'Not on Treatment' if der_anytx is 0<br>Coded as 'Palliative' if der_anytx is 1 and treatment_intent is 2<br>Coded as 'Curative' if der_anytx is 1 and treatment_intent is 1<br>Otherwise, Missing der_anytx:<br>0=No; 1=Yes; 99=Unknown Treatment_intent: 1, Curative \| 2, Palliative \| 99, Unclear or unknown |
| Most recent line of cancer treatment, including systemic and non-systemic therapies | der_txline | Untreated in last 12 months; Curative NOS; First line; Non-curative NOS; Other; Second line or greater; Unknown | |
| Hematologic malignancy indicator | der_heme | 0=No; 1=Yes | |

**Appendix 3—table 4.** Other covariates used for analysis.

| Other covariate related to cohort selection for analysis | Variable name | Covariate values | Covariate description |
|---|---|---|---|
| *Sex*<br>*(recode other/prefer not to say gender -->missing)* | *der_sex* | *Male, Female* | |
| Breast cancer | der_Breast | 0=No; 1=Yes | |
| Cancer type of second malignancy.<br>If the patient has more than two malignancies, please select the second-most recently diagnosed cancer type. If unknown or unclear, please specify in the free text box below | cancer_type_2[1] | '' indicates no second malignancy | |
| Region of patient residence with US and ex-US collapsed | der_region_v3 | Non-US; Other; US | |

**Appendix 3—table 5.** New covariates added (2-5-22).

| New covariate | Variable name | Covariate values | Covariate description |
|---|---|---|---|
| MBC vs non-MBC | der_metastatic | 0=No; 1=Yes; 99=Unknown | *Metastatic cancer status (only applicable to solid tumors/lymphoma)* |
| MBC site of metastasis | der_met_bone | 0=No; 1=Yes; 99=Unknown | Metastatic to bone |
| MBC site of metastasis | der_met_liver | 0=No; 1=Yes; 99=Unknown | Metastatic to liver |
| MBC site of metastasis | der_met_lung_v2 | 0=No; 1=Yes; 99=Unknown | Metastatic to lung |

# Appendix 4

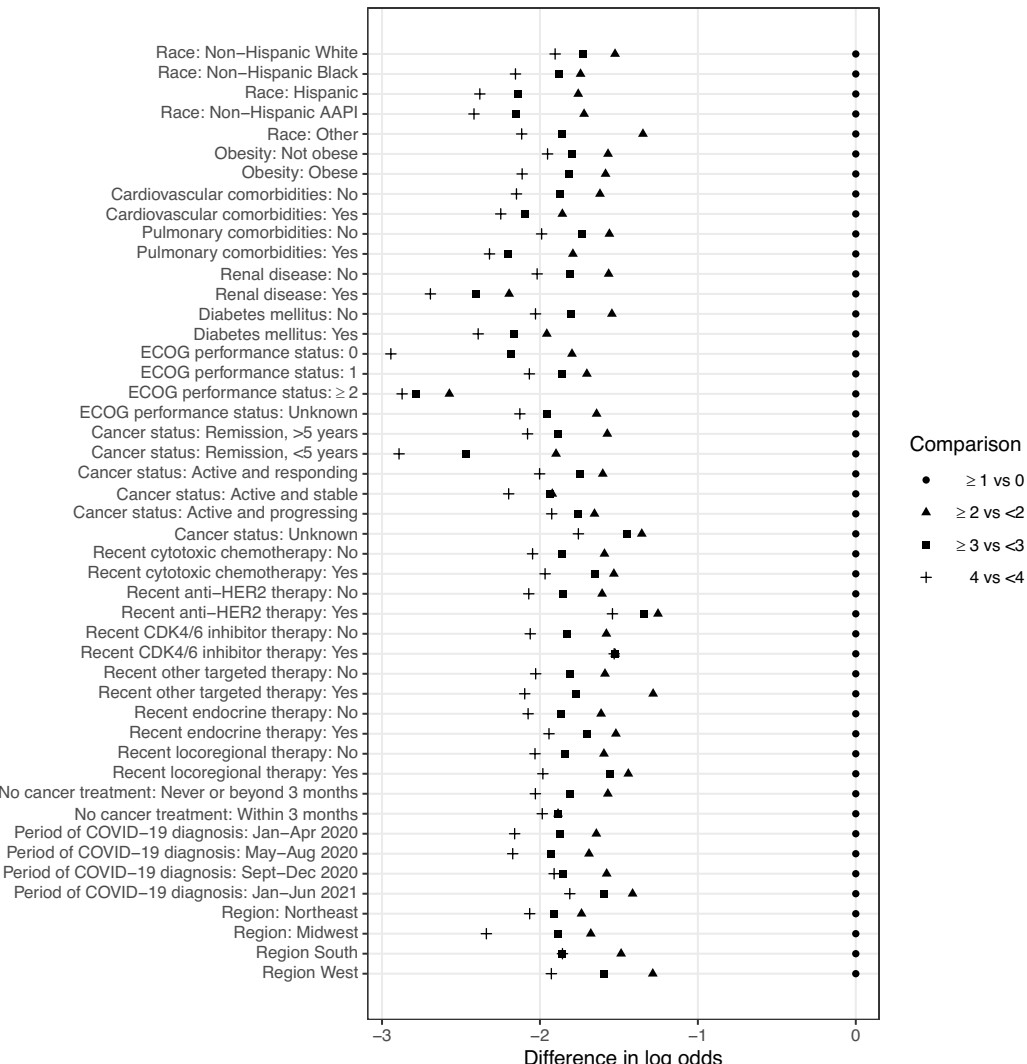

**Appendix 4—figure 1.** Represents graphical methods used to verify the proportional odds assumptions.

## Appendix 5

**Appendix 5—table 1.** Unadjusted rates of outcomes after COVID-19 diagnosis by cancer status.

| | NED >5 years | NED <5 years | Active and responding | Active and stable | Active and progressing | Missing/ unknown | Total |
|---|---|---|---|---|---|---|---|
| | n* (%) | n* (%) | n* (%) | n* (%) | n* (%) | n* (%) | n* (%) |
| *Outcomes* | | | | | | | |
| Total all-cause mortality[†] | 40 (11) | 12 (3) | 12 (7) | 11 (7) | 37 (38) | 11 (9) | 123 (9) |
| 30-day all-cause mortality [‡] | 29 (8) | 10 (2) | 10 (6) | 4 (2) | 27 (28) | 9 (7) | 89 (6) |
| Received mechanical ventilation[†] | 20 (5) | 13 (3) | 9 (5) | 7 (4) | 12 (12) | 8 (7) | 69 (5) |
| Admitted to an intensive care unit[†] | 35 (10) | 25 (6) | 13 (8) | 8 (5) | 18 (19) | 12 (10) | 111 (8) |
| Admitted to the hospital[†] | 163 (43) | 129 (29) | 54 (31) | 57 (34) | 70 (72) | 39 (32) | 512 (37) |

*N is based on non-missing data.
[†]Included in primary ordinal COVID-19 severity outcome.
[‡]Secondary outcome.

# Appendix 6

**Appendix 6—table 1.** Baseline characteristics of female patients with MBC and COVID-19.

|  | MBC |
| --- | --- |
|  | (N=233) |
| Age, years[†] |  |
| Median [IQR] | 58.0 [49.8, 68.3] |
| Race/ethnicity |  |
| Non-Hispanic White | 107 (46%) |
| Non-Hispanic Black | 56 (24%) |
| Hispanic | 50 (21%) |
| Non-Hispanic AAPI | 10 (4%) |
| Other | 10 (4%) |
| Smoking status |  |
| Never | 162 (70%) |
| Current or former | 66 (28%) |
| Missing/unknown | 5 (2%) |
| Obesity |  |
| No | 139 (60%) |
| Yes | 93 (40%) |
| Comorbidities[‡] |  |
| Cardiovascular | 42 (18%) |
| Pulmonary | 37 (16%) |
| Renal disease | 16 (7%) |
| Diabetes mellitus | 52 (22%) |
| Missing/unknown | 3 (1%) |
| ECOG performance status |  |
| 0 | 63 (27%) |
| 1 | 84 (36%) |
| 2+ | 42 (18%) |
| Unknown | 44 (19%) |
| Missing | 0 (0%) |
| Receptor status |  |
| HR+/HER2- | 98 (42%) |
| HR+/HER2+ | 53 (23%) |
| HR-/HER2+ | 26 (11%) |
| Triple negative | 33 (14%) |
| Missing/unknown | 23 (10%) |
| Cancer status |  |
| Active and responding | 55 (24%) |
| Active and stable | 78 (33%) |

*Appendix 6—table 1 Continued on next page*

*Appendix 6—table 1 Continued*

| | MBC |
|---|---|
| Active and progressing | 74 (32%) |
| Unknown | 25 (11%) |
| Missing | 0 (0%) |
| **Metastatic sites (MBC)** | |
| Lung | 65 (28%) |
| Bone | 135 (58%) |
| Liver | 61 (26%) |
| Missing/unknown | 19 (8%) |
| **Timing of anti-cancer therapy** | |
| Never/after COVID-19 | X* |
| 0–4 weeks | 189 (81%) |
| 1–3 months | 14 (6%) |
| >3 months | 19 (8%) |
| Missing/unknown | 11 (5%)* |
| **Modality of active anti-cancer therapy[‡, §]** | |
| None | 24 (10%) |
| Cytotoxic chemotherapy | 114 (49%) |
| Targeted therapy | 115 (49%) |
| Endocrine therapy | 98 (42%) |
| Immunotherapy | 17 (7%) |
| Local (surgery/radiation) | 27 (12%) |
| Other | 6 (3%) |
| Missing/unknown | 6 (3%) |
| **Region** | |
| Northeast | 97 (42%) |
| Midwest | 44 (19%) |
| South | 34 (15%) |
| West | 56 (24%) |
| Undesignated | 2 (1%) |
| **Period of COVID-19 diagnosis** | |
| Jan-Apr 2020 | 33 (14%) |
| May-Aug 2020 | 101 (43%) |
| Sept-Dec 2020 | 52 (22%) |
| Jan-Aug 2021 | 45 (19%) |
| Missing/unknown | 2 (1%) |
| **Area of patient residence** | |
| Urban | 103 (44%) |
| Suburban | 80 (34%) |

*Appendix 6—table 1 Continued on next page*

*Appendix 6—table 1 Continued*

| | MBC |
|---|---|
| Rural | 12 (5%) |
| Missing/unknown | 38 (16%) |
| **Treatment center characteristics** | |
| Academic medical center | 43 (18%) |
| Community practice | 63 (27%) |
| Tertiary care center | 127 (55%) |
| Missing/unknown | 0 (0%) |
| **Severity of COVID-19** | |
| Mild | 126 (54%) |
| Moderate | 93 (40%) |
| Severe | 13 (6%) |
| Missing/unknown | 1 (<1%) |

*Cells combined to mask N<5 according to CCC19 low count policy.
†Age was truncated at 90 years.
‡Percentages could sum to >100% because categories are not mutually exclusive.
§Within 3 months of COVID-19 diagnosis.

**Appendix 6—table 2.** Unadjusted rates of outcomes after COVID-19 diagnosis in female patients with MBC.

| | n** (%) |
|---|---|
| **Outcomes** | |
| Total all-cause mortality* | 45 (19) |
| 30-day all-cause mortality† | 28 (12) |
| Received mechanical ventilation* | 20 (9) |
| Admitted to an intensive care unit* | 29 (12) |
| Admitted to the hospital* | 124 (53) |
| **Clinical complications** | |
| Any cardiovascular complication‡ | 48 (21) |
| Any pulmonary complication§ | 86 (37) |
| Any gastrointestinal complication¶ | 13 (6) |
| Acute kidney injury | 32 (14) |
| Multisystem organ failure | 12 (5) |
| Superimposed infection | 32 (14) |
| Sepsis | 28 (12) |
| Any bleeding | 8 (3) |
| **Interventions** | |
| Remdesivir | 35 (15) |
| Hydroxychloroquine | 25 (11) |
| Corticosteroids | 65 (29) |
| Covid Other | 45 (20) |

*Appendix 6—table 2 Continued on next page*

*Appendix 6—table 2 Continued*

| | n** (%) |
|---|---|
| Supplemental oxygen | 84 (37) |

*Included in primary ordinal COVID-19 severity outcome.

†Secondary outcome.

‡Cardiovascular complication includes hypotension, myocardial infarction, other cardiac ischemia, atrial fibrillation, ventricular fibrillation, other cardiac arrhythmia, cardiomyopathy, congestive heart failure, pulmonary embolism (PE), deep vein thrombosis (DVT), stroke, thrombosis NOS complication.

§Pulmonary complication includes respiratory failure, pneumonitis, pneumonia, acute respiratory distress syndrome (ARDS), PE, pleural effusion, empyema.

¶Gastrointestinal complication includes acute hepatic injury, ascites, bowel obstruction, bowel perforation, ileus, peritonitis.

**N is based on non-missing data.

## Appendix 7

**Appendix 7—table 1.** Adjusted associations of race factors with COVID-19 severity outcome.

| | COVID-19 severity | | |
|---|---|---|---|
| | OR (95% CI) | Point value e estimates* | Lower bound e values* |
| Race (Ref: NHW) | | | |
| Black | 1.74 (1.24–2.45) | 1.97 | 1.47 |
| Hispanic | 1.38 (0.93–2.05) | 1.63 | 1.00 |
| AAPI | 3.40 (1.70–6.79) | 3.09 | 1.93 |
| Other | 2.97 (1.71–5.17) | 2.84 | 1.94 |

*These values were calculated based on the formula for logistic regression

## Appendix 8

**Appendix 8—table 1.** Baseline characteristics of male patients with breast cancer and COVID-19.

| | |
|---|---|
| Total | 25 (100%) |
| Age, years† | |
| Median [IQR] | 67.0 [60–75] |
| Race/ethnicity | |
| NHW | 13 (52%) |
| Black | 8 (32%) |
| Hispanic | <5 (<20%) |
| AAPI | 0 (0%) |
| Other | <5 (<20%) |
| Smoking status | |
| Never | 18 (72%) |
| Current or former | 7 (28%) |
| Obesity | |
| No | 12 (48%) |
| Yes | 13 (52%) |
| Comorbidities ‡ | |
| Cardiovascular | 6 (24%) |
| Pulmonary | 5 (20%) |
| Renal disease | <5 (<20%) |
| Diabetes mellitus | 11 (44%) |
| ECOG performance status | |
| 0 | 5 (20%) |
| 1 | 10 (40%) |
| 2+ | X* |
| Unknown | 10 (40%)* |
| Receptor status | |
| HR+/HER2- | 18 (72%) |
| HR+/HER2+ | 5 (20%) |
| HR+/HER2+ | X* |
| Triple negative | 0 (0%) |
| Missing/unknown | 2 (8%)* |
| Cancer status | |
| Remission or NED, >5 years | <5 (<20%) |
| Remission or NED, <5 years | 6 (24%) |
| Active and responding | <5 (<20%) |
| Active and stable | <5 (<20%) |
| Active and progressing | 5 (20%) |

*Appendix 8—table 1 Continued on next page*

*Appendix 8—table 1 Continued*

| | |
|---|---|
| Unknown | 3 (12%) |
| **Timing of anti-cancer therapy** | |
| Never/after COVID-19 | <5 (<20%) |
| 0–4 weeks | 17 (68%) |
| 1–3 months | 0 (0%) |
| >3 months | <5 (<20%) |
| Missing/unknown | 1 (4%) |
| **Modality of active anti-cancer therapy[‡, §]** | |
| None | 7 (28%) |
| Chemotherapy | 6 (24%) |
| Targeted therapy | 6 (24%) |
| Endocrine therapy | 10 (40%) |
| Immunotherapy | 0 (0%) |
| Local (surgery/radiation) | <5 (<20%) |
| Other | 0 (0%) |
| Missing/unknown | 1 (4%) |
| **Region** | |
| Northeast | 11 (44%) |
| Midwest | <5 (<20%) |
| South | <5 (<20%) |
| West | 7 (28%) |
| Undesignated | 0 (0%) |
| **Period of COVID-19 diagnosis** | |
| Jan-Apr 2020 | 10 (40%) |
| May-Aug 2020 | 9 (36%) |
| Sept-Dec 2020 | 5 (20%) |
| **Area of patient residence** | |
| Urban | 9 (36%) |
| Suburban | 8 (32%) |
| Rural | 0 (0%) |
| Missing/unknown | 8 (32%) |
| **Severity of COVID19** | |
| Mild | 11 (44%) |
| Moderate/severe | 14 (56%) |

Variable categories with one to five cases are masked by replacing with N<5 according to CCC19 policy.

*Cells combined to mask N<5 according to CCC19 low count policy.

[†]Age was truncated at 90 years.

[‡]Percentages could sum to >100% because categories are not mutually exclusive.

[§]Within 3 months of COVID-19 diagnosis.

**Appendix 8—table 2.** Unadjusted rates of outcomes after COVID-19 diagnosis among male patients with BC.

| *Outcomes* | |
| --- | --- |
| Total all-cause mortality | 5 (20) |
| 30-day all-cause mortality | 5 (20) |
| Received mechanical ventilation | <5 (<20%) |
| Admitted to an intensive care unit | <5 (<20%) |
| Admitted to the hospital | 15 (60) |
| *Clinical complications* | |
| Any cardiovascular complication* | <5 (<20%) |
| Any pulmonary complication† | 12 (48) |
| Any gastrointestinal complication‡ | 0 (0%) |
| Acute kidney injury | <5 (<20%) |
| Multisystem organ failure | <5 (<20%) |
| Superimposed infection | <5 (<20%) |
| Sepsis | <5 (<20%) |
| Any bleeding | <5 (<20%) |
| *Interventions* | |
| Remdesivir | <5 (<20%) |
| Hydroxychloroquine | 7 (28) |
| Corticosteroids | <5 (<20%) |
| Other | 9 (36) |
| Supplemental oxygen | 12 (48) |

Variable categories with one to five cases are masked by replacing with N<5 according to CCC19 policy

*Cardiovascular complication includes hypotension, myocardial infarction, other cardiac ischemia, atrial fibrillation, ventricular fibrillation, other cardiac arrhythmia, cardiomyopathy, congestive heart failure, pulmonary embolism (PE), deep vein thrombosis (DVT), stroke, thrombosis NOS complication.

†Pulmonary complication includes respiratory failure, pneumonitis, pneumonia, acute respiratory distress syndrome (ARDS), PE, pleural effusion, empyema.

‡Gastrointestinal complication includes acute hepatic injury, ascites, bowel obstruction, bowel perforation, ileus, peritonitis.

## Appendix 9

### List of participants by institution

Alphabetical list of participants by institution that contributed at least one record to the analysis.

Bolded = site PI/co-PIs; site co-investigators are listed alphabetically by last name.

- **Balazs Halmos, MD; Amit Verma, MBBS**; Benjamin A Gartrell, MD; Sanjay Goel, MBBS; Nitin Ohri, MD; R Alejandro Sica, MD; Astha Thakkar, MD (Albert Einstein College of Medicine, Montefiore Medical Center, Bronx, NY, USA)
- **Keith E Stockerl-Goldstein, MD**; Omar Butt, MD, PhD; Jian Li Campian, MD, PhD; Mark A Fiala, MSW; Jeffrey P Henderson, MD, PhD; Ryan S Monahan, MBA; Alice Y Zhou, MD, PhD (Alvin J Siteman Cancer Center at Washington University School of Medicine and Barnes-Jewish Hospital, St. Louis, MO, USA)
- **Michael A Thompson, MD, PhD, FASCO**; Pamela Bohachek, RN, CCRC; Daniel Mundt, MD; Mitrianna Streckfuss, MPH; Eyob Tadesse, MD (Aurora Cancer Care, Advocate Aurora Health, Milwaukee, WI, USA)
- **Philip E Lammers, MD, MSCI** (Baptist Cancer Center, Memphis, TN, USA)
- **Sanjay G Revankar, MD, FIDSA** (The Barbara Ann Karmanos Cancer Institute at Wayne State University School of Medicine, Detroit, MI, USA)
- **Jaymin M Patel, MD**; Andrew J Piper-Vallillo, MD; Poorva Bindal, MBBS (Beth Israel Deaconess Medical Center, Boston, MA, USA)
- **Orestis A Panagiotou, MD, PhD**; Pamela C Egan, MD; Dimitrios Farmakiotis, MD, FACP, FIDSA; Hina Khan, MD; Adam J Olszewski, MD (Brown University and Lifespan Cancer Institute, Providence, RI, USA)
- **Arturo Loaiza-Bonilla, MD, MSEd, FACP** (Cancer Treatment Centers of America, AZ/GA/IL/OK/PA, USA)
- **Salvatore A Del Prete, MD**; Michael H Bar, MD, FACP; Anthony P Gulati, MD; KM Steve Lo, MD; Suzanne J Rose, MS, PhD, CCRC, FACRP; Jamie Stratton, MD; Paul L Weinstein, MD (Carl & Dorothy Bennett Cancer Center at Stamford Hospital, Stamford, CT, USA)
- **Robin A Buerki, MD**; Jorge A Garcia, MD, FACP (Case Comprehensive Cancer Center at Case Western Reserve University/University Hospitals, Cleveland, OH, USA)
- **Shilpa Gupta, MD**; Nathan A Pennell, MD, PhD, FASCO; Manmeet S Ahluwalia, MD, FACP; Scott J Dawsey, MD; Christopher A Lemmon, MD; Amanda Nizam, MD (Cleveland Clinic, Cleveland, OH, USA)
- **Claire Hoppenot, MD; Ang Li, MD, MS** (Dan L Duncan Comprehensive Cancer Center at Baylor College of Medicine, Houston, TX, USA)
- **Toni K Choueiri, MD**; Ziad Bakouny, MD, MSc; Jean M Connors, MD; George D Demetri, MD, FASCO; Dory A Freeman, BS; Antonio Giordano, MD, PhD; Chris Labaki, MD; Alicia K Morgans, MD, MPH; Anju Nohria, MD; Andrew L Schmidt, MD; Eliezer M Van Allen, MD; Pier Vitale Nuzzo, MD, PhD; Wenxin (Vincent) Xu, MD; Rebecca L Zon, MD (Dana-Farber Cancer Institute, Boston, MA, USA) (Dana-Farber Cancer Institute, Boston, MA, USA)
- **Susan Halabi, PhD, FASCO**; Tian Zhang, MD, MHS (Duke Cancer Institute at Duke University Medical Center, Durham, NC, USA)
- **John C Leighton Jr, MD, FACP** (Einstein Healthcare Network, Philadelphia, PA, USA)
- **Gary H Lyman, MD, MPH, FASCO, FRCP**; Jerome J Graber MD, MPH; Petros Grivas, MD, PhD; Elizabeth T Loggers, MD, PhD; Ryan C Lynch, MD; Elizabeth S Nakasone, MD, PhD; Michael T Schweizer, MD; Lisa Tachiki, MD; Shaveta Vinayak, MD, MS; Michael J Wagner, MD; Albert Yeh, MD (Fred Hutchinson Cancer Research Center/University of Washington/Seattle Cancer Care Alliance, Seattle, WA, USA)
- **Sharad Goyal, MD; Minh-Phuong Huynh-Le, MD, MAS** (George Washington University, Washington, DC, USA)
- **Lori J Rosenstein, MD** (Gundersen Health System, WI, USA)
- **Peter Paul Yu, MD, FACP, FASCO**; Jessica M Clement, MD; Ahmad Daher, MD; Mark E Dailey, MD; Rawad Elias, MD; Asha Jayaraj, MD; Emily Hsu, MD; Alvaro G. Menendez, MD; Oscar K Serrano, MD, MBA, FACS (Hartford HealthCare Cancer Institute, Hartford, CT, USA)

- **Clara Hwang, MD**; Shirish M Gadgeel, MD; Sunny RK Singh, MD (Henry Ford Cancer Institute, Henry Ford Hospital, Detroit, MI, USA)
- **Melissa K Accordino, MD, MS**; Divaya Bhutani, MD; Jessica E Hawley, MD; Dawn Hershman, MD, MS, FASCO; Gary K Schwartz, MD (Herbert Irving Comprehensive Cancer Center at Columbia University, New York, NY, USA)
- **Daniel Y Reuben, MD, MS**; Mariam Alexander, MD, PhD; Sara Matar, MD; Sarah Mushtaq, MD (Hollings Cancer Center at the Medical University of South Carolina, Charleston, SC, USA)
- **Eric H Bernicker, MD** (Houston Methodist Cancer Center, Houston, TX, USA)
- **John F Deeken, MD**; Danielle Shafer, DO (Inova Schar Cancer Institute, Fairfax, VA, USA)
- **Mark A Lewis, MD; Terence D Rhodes, MD, PhD**; David M Gill, MD; Clarke A Low, MD (Intermountain Health Care, Salt Lake City, UT, USA)
- **Sandeep H Mashru, MD**; Abdul-Hai Mansoor, MD (Kaiser Permanente Northwest, OR/WA, USA)
- **Brandon Hayes-Lattin, MD, FACP**; Aaron M Cohen, MD, MS; Shannon McWeeney, PhD; Eneida R Nemecek, MD, MS, MBA; Staci P Williamson, BS (Knight Cancer Institute at Oregon Health and Science University, Portland, OR, USA)
- **Howard A. Zaren, MD, FACS**; Stephanie J Smith, RN, MSN, OCN (Lewis Cancer & Research Pavilion @ St. Joseph's/Candler, Savannah, GA, USA)
- **Gayathri Nagaraj, MD**; Mojtaba Akhtari, MD; Eric Lau, DO; Mark E Reeves, MD, PhD (Loma Linda University Cancer Center, Loma Linda, CA, USA)
- **Stephanie Berg, DO**; Natalie Knox (Loyola University Medical Center, Maywood, IL, USA)
- **Firas H Wehbe, MD, PhD**; Jessica Altman, MD; Michael Gurley, BA; Mary F Mulcahy, MD (Lurie Cancer Center at Northwestern University, Chicago, IL, USA)
- **Eric B Durbin, DrPH, MS** (Markey Cancer Center at the University of Kentucky, Lexington, KY, USA)
- **Amit A Kulkarni, MD**; Heather H. Nelson, PhD, MPH; Zohar Sachs, MD, PhD (Masonic Cancer Center at the University of Minnesota, Minneapolis, MN, USA)
- **Rachel P Rosovsky, MD, MPH; Kerry L Reynolds, MD**; Aditya Bardia, MD; Genevieve Boland, MD, PhD, FACS; Justin F Gainor, MD; Leyre Zubiri, MD, PhD (Massachusetts General Hospital Cancer Center, Boston, MA, USA)
- **Thorvardur R Halfdanarson, MD**; Tanios S Bekaii-Saab, MD, FACP; Aakash Desai, MD, MPH; Surbhi Shah, MD; Zhuoer Xie, MD, MS (Mayo Clinic, AZ/FL/MN, USA) (Mayo Clinic, AZ/FL/MN, USA)
- **Ruben A Mesa, MD, FACP**; Mark Bonnen, MD; Daruka Mahadevan, MD, PhD; Amelie G Ramirez, DrPH, MPH; Mary Salazar, DNP, MSN, RN, ANP-BC; Dimpy P Shah, MD, PhD; Pankil K Shah, MD, MSPH (Mays Cancer Center at UT Health San Antonio MD Anderson Cancer Center, San Antonio, TX, USA)
- **Gregory J Riely, MD, PhD; Elizabeth V Robilotti MD, MPH**; Rimma Belenkaya, MA, MS; John Philip, MS (Memorial Sloan Kettering Cancer Center, New York, NY, USA)
- **Bryan Faller, MD** (Missouri Baptist Medical Center, St. Louis, MO, USA)
- **Rana R McKay, MD**; Archana Ajmera, MSN, ANP-BC, AOCNP; Sharon S Brouha, MD, MPH; Angelo Cabal, BS; Sharon Choi, MD, PhD; Albert Hsiao, MD, PhD; Jun Yang Jiang, MD; Seth Kligerman, MD; Taylor K Nonato; Erin G Reid, MD (Moores Comprehensive Cancer Center at the University of California, San Diego, La Jolla, CA, USA)
- **Lisa B Weissmann, MD**; Chinmay Jani, MD; Carey C. Thomson, MD, FCCP, MPH (Mount Auburn Hospital, Cambridge, MA, USA)
- **Jeanna Knoble, MD**; Mary Grace Glace, RN; Cameron Rink, PhD, MBA; Karen Stauffer, RN; Rosemary Zacks, RN (Mount Carmel Health System, Columbus, OH, USA)
- **Sibel Blau, MD** (Northwest Medical Specialties, Tacoma, WA, USA)
- **Daniel G Stover, MD**; Daniel Addison, MD; James L Chen, MD; Margaret E Gatti-Mays, MD; Sachin R Jhawar, MD; Vidhya Karivedu, MBBS; Joshua D Palmer, MD; Sarah Wall, MD; Nicole O Williams, MD (The Ohio State University Comprehensive Cancer Center, Columbus, OH, USA)
- **Monika Joshi, MD, MRCP**; Hyma V Polimera, MD; Lauren D Pomerantz; Marc A Rovito, MD, FACP (Penn State Health/Penn State Cancer Institute/St. Joseph Cancer Center, PA, USA)

- **Elizabeth A Griffiths, MD**; Amro Elshoury, MBBCh (Roswell Park Comprehensive Cancer Center, Buffalo, NY, USA)
- **Salma K Jabbour, MD**; Christian F Misdary, MD; Mansi R Shah, MD (Rutgers Cancer Institute of New Jersey at Rutgers Biomedical and Health Sciences, New Brunswick, NJ, USA)
- **Babar Bashir, MD, MS**; Christopher McNair, PhD; Sana Z Mahmood, BA, BS; Vasil Mico, BS; Andrea Verghese Rivera, MD (Sidney Kimmel Cancer Center at Thomas Jefferson University, Philadelphia, PA, USA)
- **Sumit A Shah, MD, MPH**; Elwyn C Cabebe, MD; Michael J Glover, MD; Alokkumar Jha, PhD; Ali Raza Khaki, MD; Lidia Schapira, MD, FASCO; Julie Tsu-Yu Wu, MD, PhD (Stanford Cancer Institute at Stanford University, Palo Alto, CA, USA)
- **Suki Subbiah, MD** (Stanley S Scott Cancer Center at LSU Health Sciences Center, New Orleans, LA, USA)
- **Daniel B Flora, MD, PharmD**; Goetz Kloecker, MD; Barbara B Logan, MS; Chaitanya Mandapakala, MD (St. Elizabeth Healthcare, Edgewood, KY, USA)
- **Gilberto de Lima Lopes Jr., MD, MBA, FAMS, FASCO** (Sylvester Comprehensive Cancer Center at the University of Miami Miller School of Medicine, Miami, FL, USA)
- **Karen Russell, MD, FACP**; Brittany Stith, RN, BSN, OCN, CCRP (Tallahassee Memorial Healthcare, Tallahassee, FL, USA)
- **Natasha C Edwin, MD**; Melissa Smits, APC (ThedaCare Cancer Care, Appleton, WI, USA)
- **David D Chism, MD**; Susie Owenby, RN, CCRP (Thompson Cancer Survival Center, Knoxville, TN, USA)
- **Deborah B Doroshow, MD, PhD**; Matthew D Galsky, MD; Michael Wotman, MD (Tisch Cancer Institute at the Icahn School of Medicine at Mount Sinai, New York, NY, USA)
- **Julie C Fu, MD**; Alyson Fazio, APRN-BC; Kathryn E Huber, MD; Mark H Sueyoshi, MD (Tufts Medical Center Cancer Center, Boston and Stoneham, MA, USA)
- **Jonathan Riess, MD, MS**; Kanishka G Patel, MD (UC Davis Comprehensive Cancer Center at the University of California at Davis, CA, USA)
- **Vadim S Koshkin, MD**; Hala T Borno, MD; Daniel H Kwon, MD; Eric J Small, MD; Sylvia Zhang, MS (UCSF Helen Diller Family Comprehensive Cancer Center at the University of California at San Francisco, CA, USA)
- **Samuel M Rubinstein, MD; William A Wood, MD, MPH**; Christopher Jensen, MD (UNC Lineberger Comprehensive Cancer Center, Chapel Hill, NC, USA)
- **Trisha M Wise-Draper, MD, PhD**; Syed A Ahmad, MD, FACS; Punita Grover, MD; Shuchi Gulati, MD; Jordan Kharofa, MD; Tahir Latif, MBBS, MBA; Michelle Marcum, MS; Hira G Shaikh; MD (University of Cincinnati Cancer Center, Cincinnati, OH, USA)
- **Daniel W Bowles, MD**; Christoper L Geiger, MD (University of Colorado Cancer Center, Aurora, CO, USA)
- **Merry-Jennifer Markham, MD, FACP, FASCO**; Atlantis D Russ, MD, PhD; Haneen Saker, MD (University of Florida Health Cancer Center, Gainesville, FL, USA)
- **Jared D Acoba, MD**; Young Soo Rho, MD, CM (University of Hawai'i Cancer Center, Honolulu, HI, USA)
- **Lawrence E Feldman, MD; Kent F Hoskins, MD**; Gerald Gantt Jr., MD; Li C Liu, PhD; Mahir Khan, MD; Ryan H Nguyen, DO; Mary Pasquinelli, APN, DNP; Candice Schwartz, MD; Neeta K Venepalli, MD, MBA (University of Illinois Hospital & Health Sciences System, Chicago, IL, USA)
- **Praveen Vikas, MD** (University of Iowa Holden Comprehensive Cancer Center, Iowa City, IA, USA)
- **Elizabeth Wulff-Burchfield, MD**; Anup Kasi MD, MPH (The University of Kansas Cancer Center, Kansas City, KS, USA)
- **Christopher R Friese, PhD, RN, AOCN, FAAN; Leslie A Fecher, MD** (University of Michigan Rogel Cancer Center, Ann Arbor, MI, USA)
- **Blanche H Mavromatis, MD**; Ragneel R Bijjula, MD; Qamar U Zaman, MD (UPMC Western Maryland, Cumberland, MD, USA)
- **Jeremy L Warner, MD, MS, FAMIA, FASCO**; Alaina J Brown, MD, MPH; Alicia Beeghly-Fadiel, PhD; Alex Cheng, PhD; Sarah Croessmann, PhD; Elizabeth J Davis, MD; Stephany N Duda, PhD, MS; Kyle T Enriquez, MSc BS; Benjamin French, PhD; Erin A Gillaspie, MD, MPH;

Daniel Hausrath, MD; Cassandra Hennessy, MS; Chih-Yuan Hsu, PhD; Douglas B Johnson, MD, MSCI; Xuanyi Li, BA; Sanjay Mishra, MS, PhD; Sonya A Reid, MD, MPH; Brian I Rini, MD, FACP, FASCO; Yu Shyr, PhD; David A Slosky, MD; Carmen C Solorzano, MD, FACS; Tianyi Sun, MS; Matthew D Tucker, MD; Karen Vega-Luna, MA; Lucy L Wang, BA (Vanderbilt-Ingram Cancer Center at Vanderbilt University Medical Center, Nashville, TN, USA)

- **David M Aboulafia, MD**; Brett A Schroeder, MD (Virginia Mason Cancer Institute, Seattle, WA, USA)
- **Matthew Puc, MD**; Theresa M Carducci, MSN, RN, CCRP; Karen J Goldsmith, BSN, RN; Susan Van Loon, RN, CTR, CCRP (Virtua Health, Marlton, NJ, USA)
- **Umit Topaloglu, PhD, FAMIA**; Saif I Alimohamed, MD (Wake Forest Baptist Comprehensive Cancer Center, Winston-Salem, NC, USA)
- **Robert L Rice, MD, PhD** (WellSpan Health, York, PA, USA)
- **Prakash Peddi, MD; Lane R Rosen, MD**; Briana Barrow McCollough, BSc, CCRC (Willis-Knighton Cancer Center, Shreveport, LA, USA)
- **Mehmet A Bilen, MD**; Cecilia A Castellano; Deepak Ravindranathan, MD, MS (Winship Cancer Institute of Emory University, Atlanta, GA, USA)
- **Navid Hafez, MD, MPH**; Roy Herbst, MD, PhD; Patricia LoRusso, DO, PhD; Maryam B Lustberg, MD, MPH; Tyler Masters, MS; Catherine Stratton, BA (Yale Cancer Center at Yale University School of Medicine, New Haven, CT, USA) (Yale Cancer Center at Yale University School of Medicine, New Haven, CT, USA)

