## [Editor Report]

These data offer novel and compelling information that could impact treatment decision-making for breast cancer patients, and the development of this registry contributes a valuable resource for future research, including and beyond breast cancer. It is anticipated that this study is the first of multiple publications that leverage this important data infrastructure.

---

## [Decision Letter]

**Decision letter after peer review:**

Thank you for submitting your article "Clinical Characteristics, Racial Inequities, and Outcomes in Patients with Breast Cancer and COVID-19: A COVID-19 and Cancer Consortium (CCC19) Cohort Study" for consideration by *eLife*. Your article has been reviewed by one peer reviewer who is a member of our Board of Reviewing Editors, and the evaluation has been overseen by a Senior Editor. The reviewer has opted to remain anonymous.

The reviewer and the editors have discussed the critique, and the Reviewing Editor has drafted this to help you prepare a revised submission.

Essential revisions:

1) Please address the impact that "long COVID" might have on study findings and implications.

2) Please discuss how this registry will be used to continue to monitor the long-term impact of COVID-19 infection on cancer outcomes, including but not limited to breast cancer.

*Reviewer #1 (Recommendations for the authors):*

The current study leverages a unique and powerful consortium resource--the "COVID-19 and Cancer Consortium (CCC19)" registry. This retrospective cohort study is a key study strength and has over 120 member institutions allowed researchers to examine the impact of cancer factors on COVID-19 infection severity.

The data provided are among the most robust data available to determine the associations between factors including prognostic factors, racial disparities, interventions, complications, and cancer treatment on COVID-19 severity. This study specifically focuses on adult women with either current breast cancer (BC) or a history of BC, as well as a confirmed diagnosis of COVID-19 from this CCC19 registry (n=1383 breast cancer cases out of 12,034 cancer cases total---an exploratory analysis of 25 male breast cancer cases is also described, separately).

The authors reveal several key findings, including that obesity was not observed to impact COVID-19 among these women. The study also provides compelling data in showing no association between anti-cancer treatments on COVID-19 severity. These findings have clear implications for treatment counseling and decision making for women with breast cancer who become COVID-19 infected.

Among study weaknesses and areas for alternative considerations, the follow-up period is still relatively short, given the recency of the pandemic. The authors should consider the possibility of "long COVID" in these patients and treatment effectiveness

Overall the study contributes confirmatory and novel information, utilizing a critical national registry effort. This registry can and should be used to continue to monitor the long term impact of COVID-19 infection on cancer outcomes, including but not limited to breast cancer.

These initial findings are quite compelling though the investigators should consider examining vaccination status and timing, booster status and timing, and longer-term follow up in the cohort. This is especially important in light of the possibility of "long COVID" for cancer patients.

---

## [Author Response]

Essential revisions:1) Please address the impact that "long COVID" might have on study findings and implications.

We agree with the reviewer that long COVID or post-acute sequelae of COVID (PASC) in patients with cancer is a highly relevant public health problem and needs to be studied further. Long COVID was beyond the scope of our study aims since we focused only on outcomes of acute infection and presented these findings in the manuscript. While this study did not specifically look into long COVID, we conducted a separate preliminary analysis to examine PASC in patients with cancer using data from CCC19.* Patients with underlying comorbidities, worse ECOG PS, and more severe acute SARS-CoV-2 infection had higher rates of PASC and suffered more severe complications, and incurred worse outcomes; however, long-term follow-up with granular data in a larger sample are needed to make a conclusive statement about the impact of long COVID in patients with any type of cancer.

Reference: ASCO abstract for LongCOVID: DOI: 10.1200/JCO.2022.40.16_suppl.e18746 Journal of Clinical Oncology 40, no. 16_suppl (June 01, 2022) e18746-e18746.

We have added the following sentence in the discussion/limitation section, “Given the largely unknown long-term impact of this novel virus, systematic examination of the post-acute sequelae of COVID-19 in patients with breast and other cancer subtypes is warranted.”

2) Please discuss how this registry will be used to continue to monitor the long-term impact of COVID-19 infection on cancer outcomes, including but not limited to breast cancer.

We are still collecting follow-up data on existing cases and will continue to monitor the long-term impact of COVID-19 infection on cancer outcomes, including breast cancer. We are continually encouraging sites to keep contributing follow-up details, so that the long-term impact of COVID-19 infection on cancer outcomes can be systematically monitored.

Reviewer #1 (Recommendations for the authors):The current study leverages a unique and powerful consortium resource--the "COVID-19 and Cancer Consortium (CCC19)" registry. This retrospective cohort study is a key study strength and has over 120 member institutions allowed researchers to examine the impact of cancer factors on COVID-19 infection severity.The data provided are among the most robust data available to determine the associations between factors including prognostic factors, racial disparities, interventions, complications, and cancer treatment on COVID-19 severity. This study specifically focuses on adult women with either current breast cancer (BC) or a history of BC, as well as a confirmed diagnosis of COVID-19 from this CCC19 registry (n=1383 breast cancer cases out of 12,034 cancer cases total---an exploratory analysis of 25 male breast cancer cases is also described, separately).The authors reveal several key findings, including that obesity was not observed to impact COVID-19 among these women. The study also provides compelling data in showing no association between anti-cancer treatments on COVID-19 severity. These findings have clear implications for treatment counseling and decision making for women with breast cancer who become COVID-19 infected.

We are grateful for the encouraging response by the reviewer. We agree that this is an important study with regards to its unique and vulnerable patient population facing an unprecedented crisis.

Among study weaknesses and areas for alternative considerations, the follow-up period is still relatively short, given the recency of the pandemic. The authors should consider the possibility of "long COVID" in these patients and treatment effectivenessOverall the study contributes confirmatory and novel information, utilizing a critical national registry effort. This registry can and should be used to continue to monitor the long term impact of COVID-19 infection on cancer outcomes, including but not limited to breast cancer.

We agree with the reviewer. Due to the complete lack of information on the impact of acute infection in patients with Breast cancer, this study was urgently warranted. However, although out of scope for the current study, we acknowledge that the long-term impact of this novel virus in patients with cancer is largely unknown and this national registry should continue to follow-up with this cohort of patients.

These initial findings are quite compelling though the investigators should consider examining vaccination status and timing, booster status and timing, and longer-term follow up in the cohort. This is especially important in light of the possibility of "long COVID" for cancer patients.

Thank you for the thoughtful recommendation, and we agree with the reviewer. Vaccination status was not part of this study as vaccines were not available during the predominant time frame for this cohort and we have mentioned this in the Discussion section as a limitation. We also added a sentence mentioning the importance of studying long COVID in patients with cancer. “Given the largely unknown long-term impact of this novel virus, systematic examination of the post-acute sequelae of COVID-19 in patients with breast and other cancer subtypes is warranted.”